# AutoMat: Enabling Automated Crystal Structure Reconstruction from Microscopy via Agentic Tool Use

## Abstract

Machine learning models for interatomic potentials and force fields require high-quality structural data, yet experimental crystal structures remain limited, creating a critical gap between computational simulations and real structures. While atomic-resolution electron microscopy can provide valuable images, converting these into simulation-ready structures is time-consuming and error-prone. We present **AutoMat**, an agent-based framework that directly converts Scanning Transmission Electron Microscopy (STEM) images to atomistic crystal structures for property prediction. The framework adaptively calls tools, including pattern-adaptive denoising, physics-guided template retrieval, symmetry-aware atomic reconstruction, and property prediction, enabling closed-loop reasoning from "image $\rightarrow$ structure $\rightarrow$ property." For systematic evaluation, we introduce **STEM2Mat-Bench**, a benchmark dataset containing 450+ annotated samples. Performance is assessed using lattice root-mean-square deviation (RMSD), formation energy mean absolute error (MAE), and structure matching accuracy. Results demonstrate that AutoMat outperforms existing approaches including SOTA models, specialized domain tools, and closed-source multimodal large models. This work establishes a direct pathway from microscopic characterization to atomic-scale modeling, addressing a fundamental challenge in materials science.

## 1 Introduction

Machine learning interatomic potentials and force fields now approach *ab initio* accuracy in predicting atomic energies and forces (Yang et al., 2024; Batatia et al., 2024; Gasteiger et al., 2021; Liao et al., 2024; Lu et al., 2023) but remain limited by experimental validated crystal structures. Meanwhile, Scanning Transmission Electron Microscopy (STEM) can image atoms at sub-ångström resolution (Zhang et al., 2018; 2023; 2020; Ortalan et al., 2010), yet translating into quantitative crystal structures still relies on expert-driven, time-consuming annotation. This creates a gap between structural representation and theoretical validation in materials science (Kalinin et al., 2022; 2023).

Although recent advances in STEM image analysis have shown promise, most existing studies focus on individual components like denoising (Yang et al., 2025; Lin et al., 2020; 2021; Wang et al., 2020), atom localization (Ziatdinov et al., 2017; Borshon et al., 2024; Eliasson & Erni, 2024), reconstruction (De Backer et al., 2022; Stoops et al., 2025; Huang et al., 2025) and phase classification (Li et al., 2021; Maksov et al., 2019). These approaches remain fragmented and are not integrated into end-to-end system. Conventional image denoising techniques suppress noise and improve contrast at the pixel level but cannot yield periodic or chemically meaningful crystal structures (Gambini et al., 2023; Ihara et al., 2022; Wang et al., 2020; Yang et al., 2025). Atomic detection models can localize atomic peaks but cannot infer complete lattices[1] or identify atomic species. General-purpose multimodal models like GPT-4.1mini (Achiam et al., 2023) and Qwen2.5-VL (Bai et al., 2025) exhibit basic image understanding but lack the ability to produce simulation-ready formats such as crystallographic file (CIF). Even domain-specific tools like AtomAI (Ziatdinov et al., 2022) and the newly released domain SOTA model MicroscopyGPT (Choudhary, 2025) can only predict atomic coordinates and structural descriptions for minimal systems, without supporting

---

[1] In crystallography, a lattice refers to the periodic arrangement of atoms in a crystal structure.

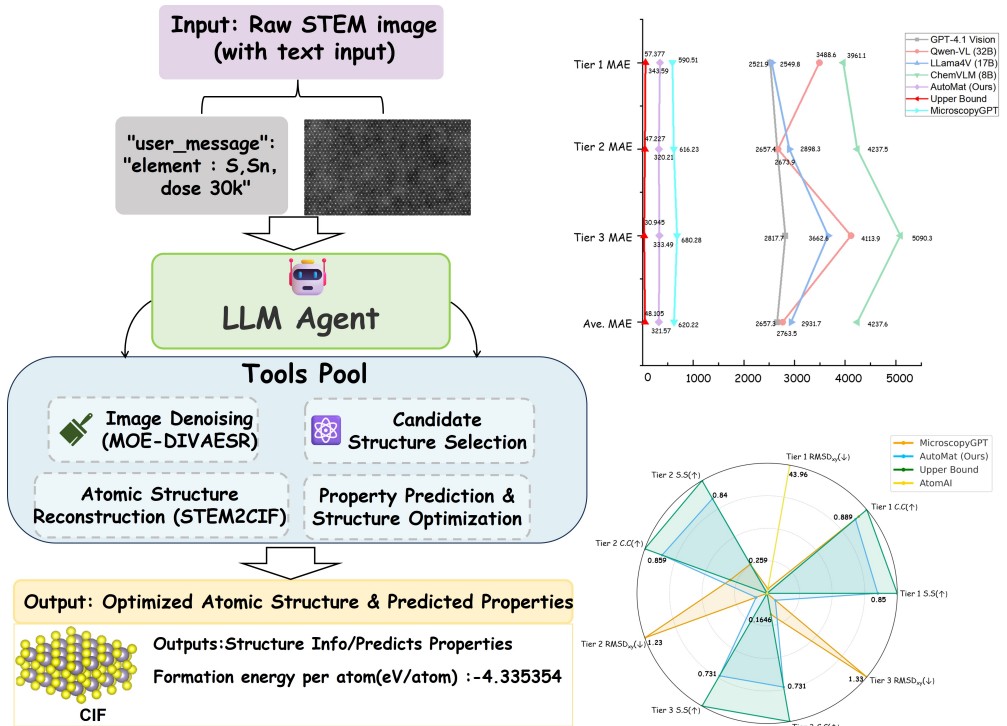

**Figure 1: Overview of AutoMat.** The left part shows an LLM agent how to manage four modules—pattern-adaptive denoising, template selection, atomic reconstruction (STEM2CIF), and ML-based property prediction. The right panel presents a line chart and radar plot comparing different models in terms of energy and structural errors.

complete CIF structure reconstruction or property prediction. Meanwhile, public datasets (e.g., JARVIS-STM) mainly target STM (Scanning Tunneling Microscopy) images, lack DFT-level energy labels, and are unsuitable for benchmarking structure–property pipelines. As a result, the field lacks a fully automated end-to-end system that can convert raw STEM images into reconstructed structures and simulated properties, along with the standardized benchmark for comprehensive evaluation.

To address this gap, as shown in Fig. 1, we introduce **AutoMat**, the first agent-based framework linking STEM imaging with atomistic simulation, along with a benchmark tailored to the task. At the core of AutoMat is an intelligent agent equipped with a suite of four powerful modular tools, which it can dynamically orchestrate to solve the task: 1) **Pattern-Adaptive STEM Image denoising**: We apply a pattern-adaptive mixture-of-experts network, MOE-Denoising Inference Variational Autoencoder Super-Resolution (Yang et al., 2025)(MOE-DIVAESR), to denoise and enhance raw iDPC-STEM[2] images. The ResNet-18-based gating network elects the most suitable expert network for each input image based on its estimated noise level, enabling joint denoising, inpainting, and super-resolution. 2) **Physics-Guided Template Retrieval**: Enhanced images are matched to a large-scale library of simulated STEM projections. Top candidate structures are retrieved using pixel similarity and filtered by elemental contrast patterns to produce strong structural priors. 3) **Symmetry-Constrained Structure Reconstruction**: Atomic peaks are detected via unsupervised clustering. We fit the lattice under symmetry constraints, assign atomic species based on the candidate, and generate the standard CIF file representing the periodic crystal structure. 4) **Energy Evaluation via Machine-Learned Potential**: The reconstructed structure is relaxed using the pretrained MatterSim potential to predict formation energy. The language agent autonomously orchestrates the workflow, planning tool execution and adaptively retrying failed steps based on quality checks.

To support rigorous evaluation, we curated 2,143 high-quality monolayer structures from C2DB (Haastrup et al., 2018), Materials Project (Jain et al., 2013), and OpenCrystal (Vaitkus et al., 2023), and

---

[2]iDPC-STEM stands for integrated Differential Phase Contrast Scanning Transmission Electron Microscopy, a technique that enhances light element contrast in atomic-resolution imaging.

simulated their corresponding iDPC-STEM images with abTEM (Madsen & Susi, 2020). From this pool we selected 450 representative image–structure pairs, which constitute our **STEM2Mat** benchmark and are used for all subsequent evaluations. Our evaluation metrics include projected lattice RMSD, formation energy MAE, and atom-wise structure matching success rate. Additionally, we introduce three fine-grained indicators to assess reconstruction quality (e.g., atomic recall/precision), robustness across noise levels, and computational efficiency. On this benchmark, AutoMat achieves a projected RMSD of $0.11 \pm 0.03$ Å, energy MAE below 350 meV/atom, and an atomic correspondence total success rate of 83.2%, outperforming GPT-4.1-mini, Qwen-VL, LLama4V, ChemVLM (Li et al., 2025), MicroscopyGPT and AtomAI by an order of magnitude. These results demonstrate AutoMat as a reproducible and accurate end-to-end solution for microscopy-driven materials modeling.

The contributions of this paper are summarized as follows:

• **Agent-based Autonomous Orchestration**. AutoMat proposes an intelligent agent that selectively orchestrates capabilities for denoising, template retrieval, reconstruction, and relaxation, dynamically charting a path from raw STEM images to material property predictions.

• **Algorithmic advancements**. We design MOE-DIVAESR as a pattern-adaptive denoiser for diverse STEM images, enabling efficient noise reduction, defect correction, and detail enhancement. We also develop STEM2CIF to reconstruct crystal structures by identifying the minimal repeating unit using symmetry heuristics and physical constraints, then converting the result into standard CIF format.

• **STEM2Mat benchmark & evaluation suite**. We release a dataset of 2,143 distinct supercell structures used to simulate large-field STEM images. In addition, we provide a 450-case test split with DFT-calculated energies and establish unified metrics to evaluate lattice reconstruction accuracy, energy fidelity, robustness, and computational efficiency.

• **State-of-the-art results with open access**. AutoMat outperforms leading large multimodal models by an order of magnitude. All code, data, and evaluation tools are publicly released to support reproducibility and future development.

## 2 RELATED WORK

**Automated Microscopy Image Analysis.** In recent years, deep learning methods have been extensively applied to electron microscopy and scanning probe microscopy data analysis (Yang et al., 2025; Lin et al., 2020; 2021; Wang et al., 2020; Ziatdinov et al., 2017; Borshon et al., 2024; De Backer et al., 2022; Stoops et al., 2025; Huang et al., 2025). Current approaches range from unsupervised defect detection to supervised atomic column identification. AtomAI (Ziatdinov et al., 2022), for example, integrates microscopy images with computational simulations, but primarily focuses on atom segmentation and identification. More recently, SciLink (Yao et al., 2025) proposes an open-source *multi-agent* framework that closes the loop between microscopy/spectroscopy experiments, literature-based novelty assessment, and theory-in-the-loop simulations, and has demonstrated impressive performance in automated defect localization and atomic-scale analysis. However, its primary focus is on serendipity-aware, high-level experiment–theory orchestration. It does not yet provide a dedicated solution for reconstructing electron-microscopy images into explicit crystal structures that can be quantitatively compared with theoretical models, nor for using such reconstructed structures as direct inputs to downstream simulations and property prediction.

**STEM Image to Structure Reconstruction.** Existing methods for reconstructing crystal structures from STEM images typically rely on multiple images, prior structural information, or are limited to single-element systems. For instance, De Backer et al (De Backer et al., 2022) employed Bayesian algorithms for 3D reconstruction, but their approach was demonstrated only for simple single-element systems. Currently, few methods can generate standard crystallographic information files (CIF) from single experimental images, especially for complex multi-element 2D crystals (Stoops et al., 2025).

**Vision-Language Models in Chemistry.** Recently, multimodal large language models (ChemVLM (Li et al., 2025), GPT-4.1mini (Achiam et al., 2023) and Qwen-VL (Bai et al., 2025)) have started to be explored in chemistry and materials science. However, these models generally lack the capability to accurately handle detailed spatial structure tasks. Existing chemical agent tools (e.g., ChemCrow (Bran et al., 2023) and Chemagents (Tang et al., 2025a)) are limited to text-based descriptions and cannot process image-based inputs, significantly restricting their applicability in microscopy-based analyses.The field's SOTA multimodal model MicroscopyGPT (Choudhary, 2025)

can generate structural descriptions from STEM images but cannot yet reconstruct CIFs or predict properties.

**Material Property Prediction Models.** Advances in machine learning-based interatomic potentials (such as MatterSim (Yang et al., 2024), M3GNet (Chen & Ong, 2022), and MACE (Batatia et al., 2023)) have significantly improved the accuracy of computational property predictions. Combining these models with experimental imaging provides a novel, digital twin-like approach for validating structural reconstructions. However, existing benchmarks predominantly evaluate models using theoretical datasets, lacking end-to-end assessments starting from experimental image inputs.

In summary, these approaches highlight progress and limitations in microscopy image analysis, structure reconstruction, multimodal modeling, and property prediction. **AutoMat addresses these gaps** by integrating an automated agent from STEM images to material property prediction and establishes the STEM2Mat benchmark for evaluating the robustness and scalability of automated material characterization.

## 3 DATASET CONSTRUCTION

### 3.1 COMPOSITION AND TAXONOMY

Our benchmark focuses on two-dimensional (2D) materials, whose atomic-scale thickness allows STEM to resolve individual columns with minimal multiple-scattering artifacts. Starting from nearly 10,000 candidate structures harvested from C2DB, Materials Project, and OpenCrystal,[3] we followed a two-stage curation process. First, automated filters removed non-stoichiometric, partially occupied, or 3D bulk entries. Second, domain experts inspected symmetry, cleavage energy, and substrate feasibility, yielding **2,143** high-confidence monolayer crystals. The collection spans six chemical families (Fig. 5): *(i)* classic 2D materials—graphene, $MoS_2$, h-BN, black phosphorus; *(ii)* emergent allotropes, e.g. silicene, borophene; *(iii)* conductive MXenes (23 distinct formulas); *(iv)* intrinsic 2D magnets such as $CrI_3$ and $Fe_3GeTe_2$; *(v)* Janus structures typified by MoSSe; and *(vi)* Ruddlesden–Popper–type 2D perovskites. Elemental diversity is broad: 67 unique elements appear, producing 76 unary 1,409 binary, and 658 ternary systems. Each structure is stored as a CIF file with validated lattice vectors and fractional atomic coordinates.

### 3.2 IMAGE SIMULATION AND DATA AUGMENTATION

To simulate realistic large-field STEM imaging conditions, we generated synthetic iDPC-STEM micrographs using the open-source `abTEM` simulation engine. For each structure, a random $12 \times 12$ to $16 \times 16$ supercell was constructed and projected at $0.1 \, \text{Å/pixel}$ resolution. Five electron-dose settings ($1$–$6 \times 10^4 \, \text{e}^-/\text{Å}^2$) and realistic lens aberrations were sampled to mimic experimental conditions. Poisson detector noise was injected to match reported signal-to-noise ratios. To study model robustness, we applied Gaussian blurring and dose-specific shot noise to simulate additional imaging imperfections. Ground-truth atomic coordinates were rendered into Gaussian "atom masks" to enable supervised training of localisation models. Each sample thus forms an image–structure–property triplet: (i) the noisy STEM projection, (ii) the corresponding ground-truth CIF, and (iii) DFT-level formation energy (along with band gap and magnetic moment, if available). We conducted principal component analysis (PCA) on structural fingerprints, which revealed clear clustering patterns. Subsequently, $k$-means clustering ($k{=}2, 6$) was used to ensure balanced train/validation/test splits across the chemical diversity of the dataset.

### 3.3 STEM2MAT BENCHMARK SPLIT AND TIERING

To construct a representative and tractable benchmark dataset for STEM-based crystal modeling, we applied stringent geometric and chemical screening criteria to the 2,143 collected 2D material structures, ensuring their suitability for monolayer imaging and end-to-end reconstruction. Specifically, we retained only structures containing no more than three distinct elements. For those with multiple elements, we required a minimum atomic-number span of ten, i.e., $\max(Z_i) - \min(Z_i) \geq 10$, to ensure sufficient imaging contrast between heavy and light atoms. To guarantee monolayer geometry, we limited the $z$-axis thickness to no more than 3 Å. Each structure's atomic coordinates were then

---

[3]All sources were queried in April 2025 using identical monolayer filters.

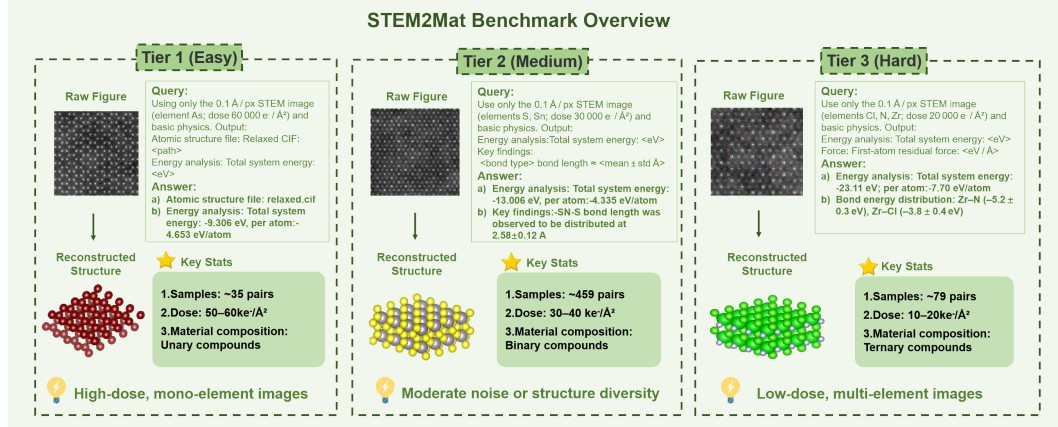

Figure 2: **Overview of the STEM2Mat Benchmark design**, illustrating the tiered classification of STEM samples by material complexity and imaging dose, which systematically stratifies reconstruction difficulty from simple unary to complex ternary compounds.

projected onto the $(x, y)$ plane, discretized onto a 1 Å grid, and evaluated for overlapping projections. Only structures with a projected duplication ratio below 10%, i.e., $\frac{\text{Number of overlapping grid points}}{\text{Total grid points}} \leq 0.1$, were retained to avoid ambiguity in atomic interpretation.

Following this multi-criteria filtering process, we retained approximately **450** well-defined, unambiguous monolayer structures—21% of the original dataset—for blind end-to-end evaluation. The remaining 1,693 samples were split into training (80%) and validation (20%) sets to support model training and tuning.

To analyze model performance as a function of task difficulty, we stratified the test set into three tiers based on material composition and imaging noise (see Fig. 2):

• **Tier 1** – Unary materials acquired under high electron doses ($5$–$6 \times 10^4$ e$^-$/Å$^2$); these images are high contrast, low noise, and represent the easiest cases (35 samples).

• **Tier 2** – Binary materials or moderate electron dose conditions ($3$–$4 \times 10^4$ e$^-$/Å$^2$); these represent intermediate complexity in both noise level and atomic diversity (456 samples).

• **Tier 3** – Ternary compounds imaged at low dose ($1$–$2 \times 10^4$ e$^-$/Å$^2$); these samples exhibit high noise, complex contrast, and are the most challenging to reconstruct (79 samples).

While the tier sizes are imbalanced, this hierarchical structure reveals a clear gradient of reconstruction difficulty, establishing a principled ladder for evaluating robustness and scalability. With the STEM2Mat-Bench tiers defined, we now specify the quantitative metrics used throughout the paper. For detailed definitions and formulae of the evaluation metrics, see evaluation metrics 3.4.

## 3.4 EVALUATION METRICS

To compare methods reproducibly, we report two *primary* metrics—energy error and lattice error—and two *holistic* metrics that reflect chemical and structural validity.

**Mean Absolute Error (MAE).** Average difference between predicted and DFT formation energies:

$$\text{MAE} = \frac{1}{N} \sum_{i=1}^{N} \left| E_i^{\text{pred}} - E_i^{\text{ref}} \right| \; [\text{meV/atom}] \tag{1}$$

**Projected Lattice RMSD.** Deviation of in-plane lattice constants:

$$\text{RMSD}_{xy} = \sqrt{\tfrac{1}{2} \left[ (a^{\text{pred}} - a^{\text{ref}})^2 + (b^{\text{pred}} - b^{\text{ref}})^2 \right]} \tag{2}$$

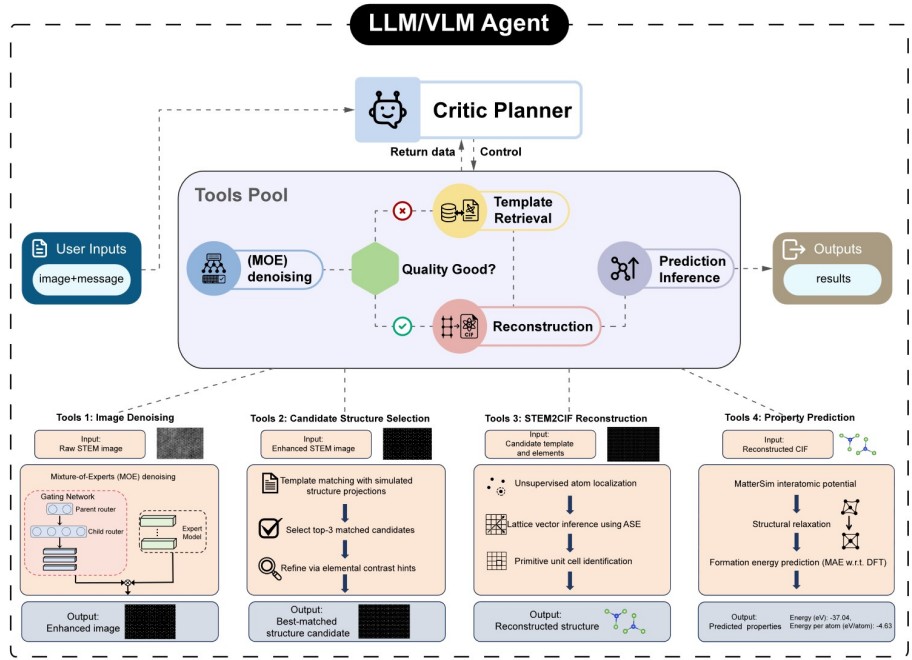

Figure 3: **AutoMat's LLM agent orchestrates four stages**—denoising, template matching, structure reconstruction, and property prediction—from STEM image to relaxed crystal with properties.

**Composition Correctness (C.C.).**  1 if elemental types and counts match the reference; 0 otherwise.

**Structure Success Rate (S.S.).**  A prediction is successful when it satisfies both chemistry and geometry. First define 2-D spatial similarity

$$S_{\text{spatial}} = \exp\left(-\text{MSE}_{2\text{D}}\right), \tag{3}$$

where $\text{MSE}_{2\text{D}}$ is the mean-squared error of projected atomic positions after optimal element-wise matching. Then

$$\text{S.S.} = \frac{1}{N} \sum_{i=1}^{N} \mathbb{I}\left[S_{\text{spatial}}^{(i)} \geq 0.8\right] \times 100\%. \tag{4}$$

## 4 OVERVIEW OF AUTOMAT

### 4.1 LIMITATIONS OF EXISTING LARGE MULTIMODAL MODELS AND DOMAIN TOOLS

Most current large multimodal models (e.g., GPT-4.1mini, LLama4V, Qwen2.5-VL, ChemVLM) focus on general-purpose image understanding tasks such as scene recognition, OCR, and molecular structure identification, but exhibit limited capability in interpreting scientific images like electron microscopy (EM). Domain-specific chemistry agents (e.g., ChemAgent, ChemCrow) remain predominantly text-based, executing tool calls for predefined tasks without closed-loop visual reasoning or image-guided decision-making. Specialized STEM data toolkits (e.g., AtomAI) provide atomic coordinate extraction and segmentation, yet their applicability is largely restricted to single-element nanoparticles and fail with complex, multi-element STEM images. MicroscopyGPT, the current state-of-the-art multimodal model in the domain, can generate structural descriptions from user-provided STEM images but still relies on known structural coordination and cannot directly reconstruct CIFs.

To date, existing agents are unable to autonomously generate simulation-ready CIFs and predict formation energies from a single real STEM image, thus completing a generalizable structure–property pipeline. To bridge this gap, we introduce **AutoMat**, an agent framework with advanced reasoning, enabling end-to-end modeling from pixel-level inputs to material property predictions.

## 4.2 TOOLS POOL

Within AutoMat—a general and efficient framework—we employ four core tools: MOE-DIVAESR, Image Template Matching, STEM2CIF, and MatterSim, responsible for image denoising, template matching, structure reconstruction, and property prediction, respectively. Further details are provided in Appendices A.2 and A.3.

**MOE-DIVAESR** is a structure-pattern–adaptive mixture-of-experts (MoE) model for STEM image denoising. Trained on an augmented STEM dataset, it contains multiple expert networks, each specializing in a distinct structural pattern, while a gating network dynamically selects the appropriate experts based on the input image features. This design enables denoising, defect correction, and fine-detail enhancement, producing sharply resolved atomic columns.

**Image Template Matching** compares an enhanced image against a database of known structures. By combining image features with elemental information, this technique efficiently narrows down potential candidates and identifies the structure that best fits experimental data. The process is even faster and more reliable when researchers can directly match against pre-calculated or known candidate structures.

**STEM2CIF** combines the selected template with the enhanced image to reconstruct a high-fidelity atomic model. It locates atomic column positions, infers lattice parameters, and identifies atomic species. Crystallographic symmetry heuristics and physical constraints then reduce the model to its minimal repeating unit, yielding a crystallographic CIF file.

**MatterSim** is a pretrained machine-learning interatomic potential for rapid structural relaxation and property evaluation. Trained on large-scale DFT datasets, it attains near-DFT accuracy for energies and properties. Integrated with ASE, it enables structure optimization and fast estimation of quantities such as total energy, formation energy, and elastic moduli, providing a swift and accurate alternative to conventional DFT.

## 4.3 FLEXIBLE TOOL-CALLING FRAMEWORK

Building on a pool of specialized tools, **AutoMat** establishes a generalizable agentic framework for flexible tool orchestration, as illustrated in Figure 3. In AutoMat, the LLM/VLM backbone serves purely as a *text-based tool orchestrator*: it never directly "sees" raw STEM images, but instead reasons over structured outputs returned by the denoising, template-matching, and STEM2CIF modules and uses them to re-plan subsequent tool calls in a hybrid, state-dependent workflow. In practice, the input is a raw STEM image, and the agent dynamically selects appropriate tools at different steps according to the current input state, aiming to achieve accurate structural reconstruction and property prediction. Specifically, when the input contains substantial noise, such as in raw STEM images, the agent calls **MOE-DIVAESR** for denoising and structural enhancement. If a single denoising step does not meet the predefined threshold, MOE-DIVAESR can be repeatedly applied. Conversely, when the input quality is sufficiently high, the agent invokes **STEM2CIF** to localize atomic peaks, fit lattice parameters, and assign atomic species, thereby reconstructing an atomic model and producing a standard CIF file. For particularly challenging samples, if repeated attempts fail to converge, the agent resorts to **Image Template Matching** to directly retrieve candidate structures, select the most consistent template, and rapidly improve output quality. At the final step, **MatterSim** is automatically called to relax the reconstructed structure and predict properties such as total energy and formation energy. Throughout the process, the agent continuously monitors intermediate results, executing rollbacks and retries when necessary to ensure both stability and accuracy.

We adopt **DeepSeekV3** as a representative backbone. Even when driven purely by a text-based model, the framework demonstrates effective and flexible tool orchestration with competitive accuracy. Importantly, AutoMat enables robust end-to-end automation from raw images to material property insights, without depending on the capability of any single model.

## 5 EXPERIMENTS AND RESULTS

### 5.1 QUANTITATIVE EVALUATION

**Baseline Overview** To evaluate **AutoMat**, we compare it against three baselines targeting reasoning, reconstruction, and oracle performance:

Table 1: **Formation-energy MAE (meV/atom) across tiers.** "↓" indicates lower is better; "–" indicates the method does not provide an energy prediction.

| Method | Tier 1 ↓ | Tier 2 ↓ | Tier 3 ↓ | Avg. ↓ |
|---|---|---|---|---|
| GPT-4.1 Vision | 2521.9 | 2657.4 | 2817.7 | 2657.3 |
| Qwen-VL (32B) | 3488.6 | 2673.9 | 4113.9 | 2763.5 |
| LLama4V (17B) | 2549.8 | 2898.3 | 3662.6 | 2931.7 |
| ChemVLM (8B) | 3961.1 | 4237.5 | 5090.3 | 4237.6 |
| MicroscopyGPT (11B) | 590.51 | 616.23 | 680.28 | 620.22 |
| AtomAI | – | – | – | – |
| **AutoMat (Ours, DeepSeek)** | **343.59** | **320.21** | **333.49** | **321.57** |
| **AutoMat (Ours, GPT-4o)** | **341.72** | **322.05** | **334.12** | **323.25** |
| Upper Bound | 57.377 | 47.227 | 30.945 | 48.105 |

Table 2: **Structural-accuracy metrics for methods that output atomic models.** $RMSD_{xy}$↓: in-plane lattice root-mean-square deviation (lower is better); C.C.↑: composition correctness; S.S.↑: structure success rate.

| Tier | Method | $RMSD_{xy}$ (Å)↓ | C.C. (%)↑ | S.S. (%)↑ |
|---|---|---|---|---|
| | AtomAI | 43.96±0.31 | 2.70 | 0.0 |
| 1 | MicroscopyGPT | 1.56±0.72 | 92.8 | 0 |
| | **AutoMat (Ours, DeepSeek)** | **0.11±0.02** | **88.9** | **85.0** |
| | **AutoMat (Ours, GPT-4o)** | **0.11±0.02** | **89.2** | **85.3** |
| | Upper Bound | 0 (def.) | 100 | 100 |
| | AtomAI | N/A | 0.0 | 0.0 |
| 2 | MicroscopyGPT | 1.23±0.75 | 30.1 | 25.9 |
| | **AutoMat (Ours, DeepSeek)** | **0.11±0.03** | **85.9** | **84.0** |
| | **AutoMat (Ours, GPT-4o)** | **0.12±0.02** | **86.4** | **84.7** |
| | Upper Bound | 0 (def.) | 100 | 100 |
| | AtomAI | N/A | 0.0 | 0.0 |
| 3 | MicroscopyGPT | 1.33±0.92 | 16.46 | 0 |
| | **AutoMat (Ours, DeepSeek)** | **0.11±0.03** | **73.1** | **73.1** |
| | **AutoMat (Ours, GPT-4o)** | **0.11±0.02** | **72.8** | **72.8** |
| | Upper Bound | 0 (def.) | 100 | 100 |

- **Vision–Language Models (VLM).** GPT 4.1mini, Qwen2.5 VL (32B), LLama4V (17B), ChemVLM (8B) and MicroscopyGPT (11B) receive a fixed prompt, composition hints, and a STEM image to infer material properties, assessing multimodal reasoning.
- **AtomAI.** AtomAI's segmentation network detects atomic centers; relative coordinates plus image resolution are used to fit the lattice. This baseline measures reconstruction quality only.
- **Ground-Truth CIF + MLIP (Oracle Upper Bound).** The true CIF is fed directly to the MatterSim MLIP to benchmark formation-energy error under perfect-structure assumptions.

We summarize the performance of these baselines and our **AutoMat** system on the test set in Tables 1 and 2, which together cover samples across Tier 1–3 difficulty levels. As detailed in Appendix A.8, we further compare different LLM and VLM backbones as agent models on a test subset. Here we only present results using DeepSeek (LLM) and GPT-4o (VLM) as representative backbone models, but experiments show that regardless of which model is used as the backbone, AutoMat achieves comparable and consistent performance.

**Discussion of Results.** For energy prediction, **AutoMat** achieves a mean formation energy MAE of $332 \pm 12$ meV/atom, with tier-wise results of 343.59, 320.21, and 333.49 meV/atom. Although this is higher than the MLIP lower bound of 57 meV/atom, it is still far better than the multi-eV errors of vision–language models. Even compared with MicroscopyGPT—which may have been trained on data overlapping with our test set—AutoMat remains much stronger, with MicroscopyGPT reaching only about 620 meV/atom. As task difficulty increases, most baselines show higher MAE, confirming the soundness of our tiered benchmark. These results indicate that AutoMat's remaining errors come mainly from reconstruction rather than MLIP limits, and that its predicted structures are reliable for downstream property evaluation. Figure 4 shows two examples from Tier 2 and Tier 3, highlighting AutoMat's superior performance over GPT-4.1mini and LLaMA4, especially under low-dose and multi-element conditions.

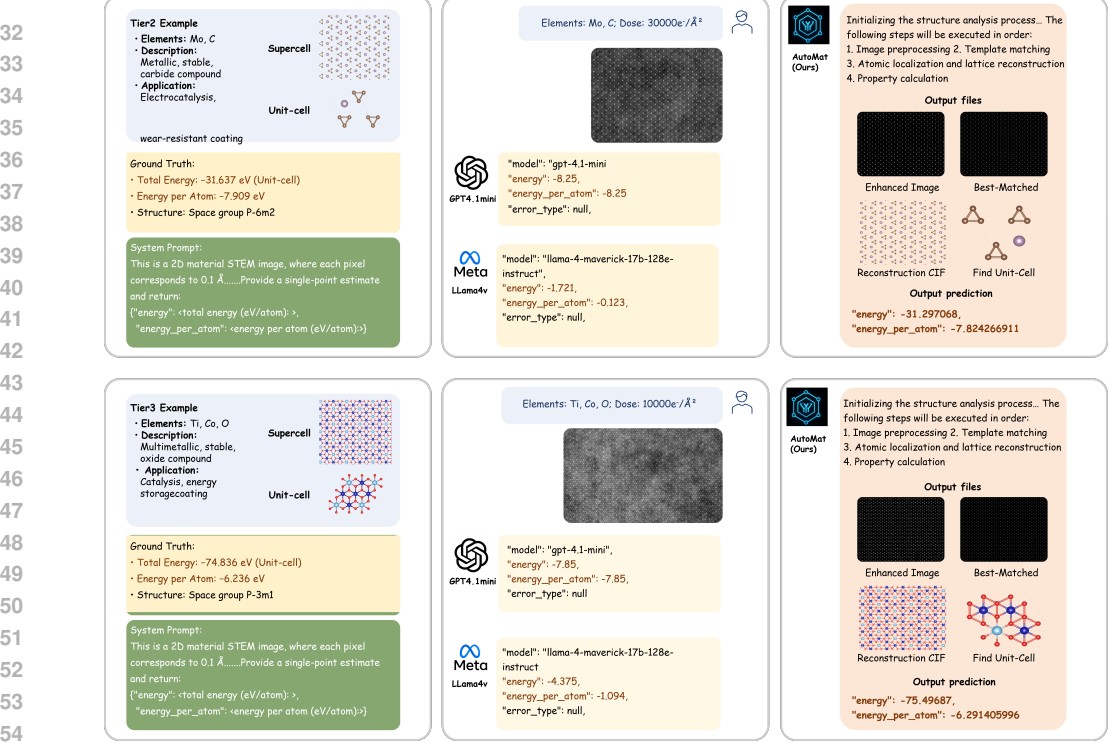

Figure 4: Case studies on Tier 2 and Tier 3 samples comparing AutoMat, GPT-4.1mini and Llama-4-Maverick: enhanced images, best-matched templates, reconstructed unit cells and predicted energies.

For structural reconstruction, AutoMat achieves an average projected $RMSD_{xy}$ of about 0.11 Å, much lower than MicroscopyGPT (1.3–1.6 Å, requiring prior coordination knowledge) and AtomAI (43–44 Å). Most deviations can be corrected through final relaxation. In composition correctness, AutoMat averages 83% across tiers (88.9%, 85.9%, 73.1%), while MicroscopyGPT performs well only in Tier 1 ($\approx$92%) but drops sharply in Tier 2–3. AtomAI, by contrast, stays below 2.7%. For structure success rate, AutoMat achieves 83.2% overall (85.0%, 84.0%, 73.1%), compared with only 25.9% for MicroscopyGPT in Tier 2 and almost zero for AtomAI.

In summary, **AutoMat** not only outperforms all baselines but also maintains strong results in challenging Tier 3 cases with multi-element compositions and low imaging doses, demonstrating robustness and generalizability across the full benchmark.

## 5.2 REAL iDPC-STEM CASE STUDY AND HUMAN-EXPERT EFFORT

**Real iDPC-STEM case.**   To complement the simulation-based benchmark, we also evaluate **AutoMat** on a real iDPC-STEM case: a ZSM-5 zeolite sample acquired on the same Cs-corrected STEM under dose-limited conditions. Without any manual tuning, the agent automatically performs denoising and structural reconstruction, and outputs a CIF structure whose projected lattice and channel framework qualitatively agree with the known crystallographic model. A concrete example is provided in Appendix A.10. This demonstrates that AutoMat has a certain ability to transfer the image–structure–property pipeline from abTEM simulations to real microscope images.

**Human-expert effort.**   To provide an intuitive comparison, we consulted a senior electron-microscopy expert in our group. Based on their experience with Tier-2–level STEM images, manually interpreting a single large-field image (1024×1024 pixels), confirming lattice parameters and elemental species, and producing a simulation-ready CIF of comparable complexity typically requires about 6–8 hours per sample. In contrast, on our benchmark hardware, **AutoMat** processes similar Tier-2 test samples in roughly 2 minutes per case. Even when targeting expert-level structural quality, this corresponds to an orders-of-magnitude improvement in throughput.

### 5.3 ERROR ANALYSIS

To better understand the failure modes of **AutoMat**, we analyzed representative failure cases across all three tiers and identified two primary types of errors (a detailed analysis with examples is provided in Appendix A.9):

**(1) Template retrieval failure (39.3%):** In these cases, AutoMat failed to retrieve the correct structure from the template database, resulting in mismatches in atomic arrangements and element types. This led to cascading errors in structure, composition, and property predictions. Incorrect atom counts further caused large energy estimation errors.

**(2) Downstream failure despite correct template (60.7%):** Even with the correct template, downstream steps failed due to projection ambiguity or elemental confusion. In 40% of these cases, atoms appeared too close in the 2D projection, and the lack of z-axis information led to poor relaxation and inaccurate energy estimates. In 20.7%, elements with similar atomic numbers (e.g., C and O) exhibited indistinguishable contrast, causing misclassification and full breakdowns in lattice fitting and CIF generation.

These findings highlight two key directions for improving **AutoMat**: (i) improving retrieval robustness via uncertainty-aware; and (ii) overcoming 2D projection limits through 3D-aware modeling Guo et al. (2023); Tang et al. (2024; 2025b) and enhanced modality integration. Together, these efforts can greatly improve structural fidelity and prediction reliability in complex systems.

## 6 CONCLUSION

We proposed **AutoMat**, an end-to-end agent system that automatically reconstructs material structures and predicts properties from STEM images. By integrating pattern-adaptive vision models, symmetry-constrained reconstruction, and LLM-driven orchestration, AutoMat achieves accurate alignment between microscopy data and atomic modeling, significantly outperforming existing methods in structural and energetic evaluation. Meanwhile, we propose STEM2Mat-Bench for this task. AutoMat advances autonomous materials research and AI-driven experimentation. Future work will focus on strengthening its role as a bridge between experimental characterization and theoretical computation.

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

# A APPENDIX

## A.1 DATASET OVERVIEW

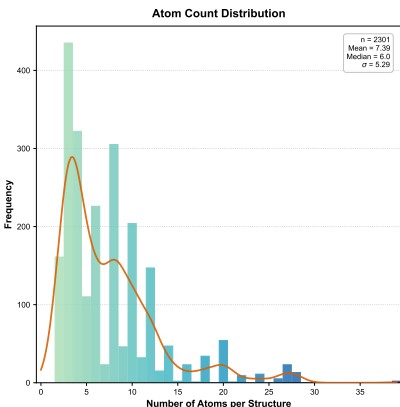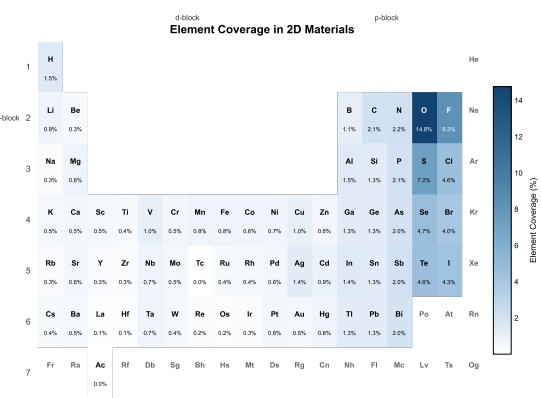

Figure 5: Distribution of per-unit-cell structure atom counts (left) and elemental coverage percentages in our curated 2D materials dataset (right).

## A.2 MOE–DIVAESR: TRAINING PIPELINE & ARCHITECTURE

### A. Training pipeline

1. **Dataset partitioning.** A total of 1,693 denoised STEM images were projected to a PCA-reduced feature space and clustered using $k$-means ($k=8$). The resulting clusters were grouped into **2 parent** and **6 child** routing categories, defining the hierarchical MoE.

2. **Prototype selection & simulation.** For each child cluster, the top 25% most representative samples (424 total) were re-simulated into iDPC-STEM projections. These images were augmented with random rotation, flipping, Poisson noise, Gaussian blur, elastic deformation, and sliding-window cropping, resulting in a total of **756,042** training patches of size $128 \times 128$.

3. **Stage 1: Router pretraining.** A ResNet-18 model was trained as a routing network for 30 epochs to predict the appropriate expert index. It achieved a top-1 routing accuracy of **99%** on the validation set. The weights were then frozen for expert training.

4. **Stage 2: Expert training.** Six DIVAESR experts (EDSR backbone with 12 residual blocks and 64 feature maps) were trained jointly under fixed routing for 300 epochs. Compared to a single DIVAESR baseline, the MoE model reduced reconstruction error by approximately 50%.

### B. Model Architecture and Objective

Each expert $E_k$ is modeled as a two-stage cascade of a denoising module and a super-resolution module:

$$E_k = g_{\phi_k}^{\text{SR}} \circ f_{\theta_k}^{\text{DIVAE}}, \qquad k = 1, \ldots, K \tag{5}$$

where $f_{\theta_k}^{\text{DIVAE}}$ is the denoising network and $g_{\phi_k}^{\text{SR}}$ is the super-resolution network.

The router $R_\psi$ produces a one-hot routing mask $\mathbf{r} \in \{0,1\}^K$ that selects expert $k^*$ for each input:

$$k^* = R_\psi(x^{\text{noisy}}) \tag{6}$$

The output of the MoE model is then computed as:

$$\hat{x}^{\text{HR}} = \sum_{k=1}^{K} r_k \cdot g_{\phi_k}^{\text{SR}}\left(f_{\theta_k}^{\text{DIVAE}}(x^{\text{noisy}})\right) \tag{7}$$

**Joint loss objective.** The total training loss for a mini-batch is given by:

$$\mathcal{L}_{\text{MoE}} = \sum_{k=1}^{K} r_k \cdot \left( \mathcal{L}_{\text{DIVAE}}^{(k)} + \mathcal{L}_{\text{SR}}^{(k)} \right) + \lambda \sum_{k=1}^{K} \bar{p}_k \log \bar{p}_k \qquad (8)$$

where:

$$\mathcal{L}_{\text{DIVAE}}^{(k)} = \frac{1}{N} \sum_{i=1}^{N} \left\| \hat{x}_i^{\text{clean}} - x_i^{\text{clean}} \right\|_2^2 + \beta \cdot \text{KL} \left[ q(z|x_i^{\text{noisy}}) \, \| \, \mathcal{N}(0, I) \right] \qquad (9)$$

is the $\beta$-VAE denoising loss, and

$$\mathcal{L}_{\text{SR}}^{(k)} = \frac{1}{N} \sum_{i=1}^{N} \left\| \hat{x}_i^{\text{HR}} - x_i^{\text{HR}} \right\|_1 \qquad (10)$$

is the L1 reconstruction loss for super-resolution. Here, $\bar{p}_k$ denotes the empirical routing frequency for expert $k$, and the KL divergence term regularizes the latent encoding.

This formulation jointly minimizes noise and detail loss through specialized expert modules. The use of KL regularization in Eq. equation 9 ensures stability in latent representation, while Eq. equation 10 preserves fine image details. The entropy term in Eq. equation 8 promotes balanced expert usage. Altogether, the MOE-DIVAESR system delivers higher-fidelity reconstruction and improved downstream performance (e.g., energy MAE) compared to single-model baselines.

### A.3 TEMPLATE MATCHING AND STEM2CIF CONVERSION

**A. Image Template Matching**

1. **Feature extraction.** Atomic centroids $\{\mathbf{p}_i\}_{i=1}^{N}$ are detected in each denoised image. An $m=32$-bin radial-distribution histogram $\mathbf{g} = (g_1, \ldots, g_m)$ is computed:

$$g_b = \frac{1}{N} \sum_{i<j} \mathbb{I}[r_{b-1} \leq \|\mathbf{p}_i - \mathbf{p}_j\| < r_b], \quad b = 1, \ldots, m, \qquad (11)$$

where $\{r_b\}$ are uniformly spaced radial bin edges, and $\mathbb{I}[\cdot]$ is the indicator function.

2. **Candidate retrieval.** All template images are similarly encoded as RDF descriptors $\{\mathbf{g}^{(t)}\}$. The cosine distance is computed as:

$$d_{\text{cos}} \left( \mathbf{g}, \mathbf{g}^{(t)} \right) = 1 - \frac{\langle \mathbf{g}, \mathbf{g}^{(t)} \rangle}{\|\mathbf{g}\| \cdot \|\mathbf{g}^{(t)}\|}, \qquad (12)$$

and the top-$k$ nearest templates ($k = 10$ by default) are retrieved.

3. **Element filtering.** Retain only templates whose elemental composition $\mathcal{E}^{(t)}$ matches that of the input:

$$\mathcal{E}^{(t)} = \mathcal{E}^{(\text{input})}. \qquad (13)$$

From the remaining candidates, select the one with the smallest $d_{\text{cos}}$.

4. **Output.** The top-1 matched template is copied to the output directory for use in downstream structure reconstruction.

Equation equation 11 encodes pairwise geometric statistics that are invariant to translation and rotation, facilitating global matching. The cosine similarity in Eq. equation 12 offers scale-invariant structural comparison. Combined with elemental filtering, this method reduces false matches in complex multi-element systems and provides high-quality priors for reconstruction.

**B. STEM2CIF Conversion**

1. **Peak localization.** Atomic column centers were localized using weighted mean-shift clustering followed by sub-pixel 2D Gaussian fitting.

2. **Lattice fitting.** An initial lattice was fitted using least-squares optimization based on known pixel resolution (0.1 Å/pixel), constrained by the space group symmetry of the matched template.

3. **Element assignment.** Atomic column intensities were compared to simulated iDPC-STEM contrast values, and the most probable element was assigned to each atomic site.

4. **CIF generation.** The reconstructed periodic structure was reduced to its primitive unit cell and exported as a standard CIF file for downstream MatterSim relaxation and property prediction.

## A.4 ABLATION STUDY

The STEM2CIF module is the only component in our system that converts microscopy images into atomic structures in the form of CIF files. Removing this module would prevent the generation of any structural outputs, which in turn makes key evaluation metrics—such as RMSD, composition correctness, or formation-energy MAE—undefined. Therefore, an ablation study that omits STEM2CIF is technically infeasible and would not yield meaningful results.

To isolate the specific contribution of STEM2CIF, we conducted an "Upper Bound" experiment (see Table 2 in the main text), where ground-truth structures were directly fed into the module, bypassing earlier steps. This experiment quantifies the module's best-case reconstruction accuracy and demonstrates its essential role in the pipeline.

To evaluate the contributions of individual modules in the AutoMat pipeline, we conduct an ablation study by selectively disabling (i) pattern-adaptive denoising (MOE-DIVAESR), and (ii) physics-guided template matching, while keeping the rest of the pipeline unchanged. We report the mean absolute error (MAE) of formation energy per atom across all three evaluation tiers(Tables 3).

Table 3: Energy per Atom MAE (meV/atom) under different ablation settings. Lower is better.

| Method | Tier 1 | Tier 2 | Tier 3 |
|---|---|---|---|
| *No Denoising (w/o MOE-DIVAESR)* | 6584 | 2616 | 938 |
| *No Template Matching* | 617 | 608 | 672 |
| **Full Pipeline (AutoMat)** | **344** | **320** | **333** |

The results show that removing MOE-DIVAESR significantly increases the error in Tier 1, indicating its critical role in enhancing low-noise images. In contrast, removing template matching severely affects Tier 3, where accurate prior guidance is essential for complex, low-dose structures. AutoMat achieves the best results across all tiers, demonstrating that both components are complementary and necessary.

## A.5 BROADER IMPACT

AutoMat offers the potential to significantly accelerate the discovery and validation of novel materials by automating the labor-intensive process of structure reconstruction from electron microscopy images. This can reduce the reliance on expert annotations and lower the barrier to entry for scientific exploration, particularly in under-resourced research settings. The open-access nature of our dataset and code also promotes transparency and reproducibility in materials science, fostering collaboration and inclusive participation.

## A.6 EXPERIMENTAL SETTINGS

**Framework.** Model training and experiment management were implemented using the PyTorch Lightning framework on a 4x NVIDIA H100 NVLink server (80GB per GPU). Hyperparameter tuning for MOE-DIVAESR required approximately 4–5 days(See Tables 4 for more configuration details.).

Table 4: Configuration of MOE-DIVAESR and related submodules

| **Gating Network** | |
| --- | --- |
| Parent Router Modules | 2 |
| Child Router Modules | 6 |
| **Latent Prior Module (DIVAESR)** | |
| Input Channels | 1 |
| Latent Dimension | 128 |
| **Super-Resolution Module (DIVAESR)** | |
| Residual Blocks | 12 |
| Feature Maps per Block | 64 |
| Activation Function | ReLU |
| Patch Size | 128 |
| Scale Factor | $\times 2$ |
| Image Channels | 1 (grayscale) |
| Precision | FP32 |
| **Optimization and Training Settings** | |
| Optimizer | Adam |
| Initial Learning Rate | 0.001 |
| Learning Rate Scheduler | Exponential decay (gamma = 0.95) |
| Weight Decay | 0.0 |
| KL Divergence Weight | 0.00025 |
| Manual Seed | 22 |
| Training Epochs | 300 |
| Training Batch Size | 128 |
| Validation Batch Size | 64 |
| Number of Workers | 8 |
| Active GPU Device | GPU ID 3 |

### A.7 MICROSCOPYGPT BASELINE

In our experiments, MicroscopyGPT is used as an off-the-shelf vision–language baseline without any additional fine-tuning on top of the released checkpoint. We rely on the model introduced by Choudhary *et al.*, which fine-tunes an 11B LLaMA-3.2–Vision model with QLoRA on a chemically and structurally diverse collection of simulated STEM images generated with AtomVision from the JARVIS-DFT-2D and C2DB two-dimensional materials databases. The training set comprises on the order of 5 000 2D materials; each example pairs a 2D STEM projection (mainly along the (001) direction) with a structured textual description containing lattice parameters, lattice angles, element types, and fractional atomic coordinates, and the model is evaluated on a 10% held-out test split using distributional metrics (KLD, EMD) and coordinate RMSE for atomic positions.

Because the MicroscopyGPT training data already cover a large class of simulated 2D monolayers similar to those in our benchmark, we deliberately do *not* further fine-tune the model on STEM2Mat-Bench. Additional fine-tuning on overlapping data could lead to overfitting or catastrophic forgetting and would make the baseline less comparable across studies. Instead, we treat MicroscopyGPT as a frozen black-box predictor and evaluate it under the same metrics as AutoMat. Concretely, for each benchmark image we (i) feed the STEM image to MicroscopyGPT with a standard structure-generation prompt, (ii) parse the textual output (lattice constants, angles, species, fractional coordinates, and Miller index) into a structured representation, and (iii) convert this representation into a CIF file using ASE and related crystallographic utilities. The resulting CIFs are then fed into the

same evaluation pipeline as AutoMat to compute RMSD, C.C., S.S., and energy-related errors. The complete inference and text-to-CIF conversion scripts will be released in our public code repository.

## A.8 ROBUSTNESS ACROSS LLM/VLM AGENTS AND RETRY MECHANISM

**Retry Mechanism.** We implement **in-context learning from environmental feedback**, where error messages and traceback logs are automatically fed into subsequent conversation turns. This enables the agent to refine function calls until either the maximum retry limit is reached or the tool call succeeds. The following example illustrates a failed attempt to call the `stem2cif_tool` due to parameter format errors, followed by an automatic correction and successful execution:

```
[TOOL_CALL] stem2cif_tool
{
  "img_path": ...,
  "elements": ["C", "Y"]
}
[TOOL_RESPONSE] stem2cif_tool
An error occurred: ValueError: Unexpected
[ANSWER]
Tool call failed due to a parameter format error.
Correct the parameters and Retry.
[TOOL_CALL] stem2cif_tool
{"img_path": ..., "elements": ["C", "Y"]}
[TOOL_RESPONSE] stem2cif_tool
{success: true, cif_path: ...}
```

This retry mechanism demonstrates the system's robustness: failures are not final but provide useful signals to guide subsequent decisions.

**Overall Sensitivity to the Choice of LLM/VLM Agent.** To address concerns that AutoMat's performance may depend on a specific backbone model (e.g., DeepSeek V3), we conducted controlled experiments replacing the LLM/VLM with Qwen3, GPT-4o, and Qwen-VL (72B)(refer to Tables 5 and Tables 6). Using a stratified random sample of 230 test cases, we observed that performance remained stable across all variants, with the average formation-energy MAE within $\pm 3\%$. This result confirms that the innovation lies in the framework design and agentic tool-use paradigm rather than reliance on a particular model.

Table 5: Formation-Energy MAE (meV/atom) Across Tiers for Different LLM and VLM Agents

| Method | Tier 1 ↓ | Tier 2 ↓ | Tier 3 ↓ | Avg. ↓ |
|---|---|---|---|---|
| Qwen-VL-agent (72B) | 261.6 | 308.5 | 446.1 | 313.84 |
| Qwen3-agent (2025-04-28) | 245.2 | 306.2 | 435.1 | 310.54 |
| GPT-4O-agent (2025-03-27) | 240.7 | 304.2 | 435.0 | 308.54 |
| DeepSeekV3-agent | 243.8 | 305.3 | 433.5 | 309.58 |

**Decision-Making Flexibility and Tool Invocation Statistics.** AutoMat functions as an agentic, end-to-end controller rather than a fixed pipeline. It dynamically decomposes user instructions into subtasks (denoising, template matching, structure reconstruction, property prediction) and adapts tool invocation depending on intermediate outcomes. If a tool fails (e.g., invalid reconstruction), the system triggers rollback-and-retry until success or termination. Moreover, the agent can skip, re-order, or re-rank templates, enabling robust performance on ill-posed problems.

To quantify this behavior, Table 7 reports detailed tool usage and retry statistics across different agents.

These results show that although success and retry rates vary across models, the overall pipeline remains stable and reliable, highlighting AutoMat's robustness and generalizability.

Table 6: Structural Accuracy Metrics for Different LLM and VLM Agents Across Tiers. RMSDxy↓: in-plane lattice RMSD; C.C.↑: composition correctness; S.S.↑: structure success rate.

| Tier | Method | RMSDxy (Å) ↓ | C.C. (%) ↑ | S.S. (%) ↑ |
|---|---|---|---|---|
| 1 | Qwen-VL (72B) | $0.12 \pm 0.02$ | 100 | 100 |
|   | Qwen3-agent | $0.12 \pm 0.01$ | 100 | 100 |
|   | GPT-4O-agent | $0.12 \pm 0.01$ | 100 | 100 |
|   | DeepSeekV3-agent | $0.11 \pm 0.02$ | 100 | 100 |
| 2 | Qwen-VL (72B) | $0.11 \pm 0.03$ | 83.1 | 79.2 |
|   | Qwen3-agent | $0.11 \pm 0.02$ | 83.3 | 79.4 |
|   | GPT-4O-agent | $0.11 \pm 0.02$ | 84.2 | 80.1 |
|   | DeepSeekV3-agent | $0.11 \pm 0.02$ | 85.9 | 79.8 |
| 3 | Qwen-VL (72B) | $0.12 \pm 0.02$ | 58.3 | 58.3 |
|   | Qwen3-agent | $0.11 \pm 0.02$ | 66.7 | 66.7 |
|   | GPT-4O-agent | $0.11 \pm 0.02$ | 66.7 | 66.7 |
|   | DeepSeekV3-agent | $0.11 \pm 0.03$ | 66.7 | 66.7 |

Table 7: Tool Invocation, Success, and Retry Statistics of Different LLM/VLM Agents in the AutoMat Toolchain.

| LLM Agent | Tool | Total Calls | Callbacks | Successes | Retries | Success Rate (%) |
|---|---|---|---|---|---|---|
| Qwen-VL (72B) | Denoising | 293,417 | 259,217 | 259,217 | 34,200 | 88.34 |
|  | Property Prediction | 165,594 | 121,153 | 68,532 | 97,062 | 41.39 |
|  | STEM2CIF | 314,001 | 252,801 | 133,282 | 180,719 | 42.45 |
|  | Template Matching | 258,987 | 205,762 | 205,762 | 53,225 | 79.45 |
| Qwen3 (2025-04-28) | Denoising | 64,929 | 58,142 | 58,142 | 6,787 | 89.55 |
|  | Property Prediction | 40,040 | 32,563 | 32,563 | 7,477 | 81.33 |
|  | STEM2CIF | 47,806 | 40,270 | 40,270 | 7,536 | 84.24 |
|  | Template Matching | 57,912 | 48,036 | 48,036 | 9,876 | 82.95 |
| GPT-4O (2025-03-27) | Denoising | 58,053 | 56,950 | 56,950 | 1,103 | 98.10 |
|  | Property Prediction | 38,705 | 36,563 | 36,563 | 2,142 | 94.46 |
|  | STEM2CIF | 46,600 | 43,507 | 43,507 | 3,093 | 93.36 |
|  | Template Matching | 59,506 | 55,951 | 55,951 | 3,555 | 94.02 |

## A.9 FAILURE ANALYSIS

**Failure Analysis.** We provide a detailed breakdown of downstream failure cases to clarify the system's limitations. Failure analysis was conducted with rollback statistics using the Qwen3 agent, with similar outcomes observed across other LLM/VLM backbones. Failures can be grouped into two major categories:

- **Template-retrieval failures (39.3%).** These arise when the correct structure is absent from the template database or visually indistinguishable within it. *Representative case:* Tier 1, `2dm-5936` (Se-oxide vs. nitride). High-Z Se columns resemble N, while light elements are suppressed in Z-contrast, leading to the selection of a P-based template and subsequent errors.

- **Downstream failures despite correct template (60.7%).** In these cases, the correct template is retrieved, but downstream reconstruction fails due to projection effects or extreme contrast imbalance. Two common sub-modes are summarized in Table 8.

In summary, this analysis highlights two important aspects. First, the indispensability of `STEM2CIF`, without which structural reconstruction and quantitative evaluation are impossible. Second, failure

Table 8: Breakdown of downstream failure sub-modes despite correct template retrieval.

| Sub-mode | Share | Core Problem | Representative Case |
|---|---|---|---|
| Projection overlap | 40% | Atoms in different $z$-layers project to the same $(x, y)$, confusing structural relaxation | Tier 2, `2dm-1014`: B/I overlap; I dominates IDPC, B is lost |
| Extreme Z-contrast | 20.7% | Heavy elements dominate contrast and mask lighter atoms, treated as noise | Tier 3, `2dm-5199`: U contrast hides O/F, leading to $> 3$ eV energy error |

modes are not random but arise from well-defined imaging and physical conditions, such as projection overlap and extreme Z-contrast. These insights provide clear directions for future work on improving robustness and addressing systematic limitations in the pipeline.

### A.10 Real STEM Image Reconstruction on ZSM-5

Due to experimental limitations, our current setup does not yet support systematic preparation and iDPC-STEM imaging of 2D materials, and it is difficult to obtain high-quality 2D samples (with both suitable morphology and stability) within a short time frame. To nevertheless provide an honest and informative evaluation of *AutoMat* on real experimental data under our present conditions, we chose ZSM-5 zeolite — a system that our group has extensively studied and can reliably synthesize — as a representative test case. Using ZSM-5 allows us to assess how well *AutoMat* generalizes from simulated iDPC-STEM images to real microscopy images in a realistic laboratory scenario.

**ZSM-5 iDPC-STEM imaging experiments.** The iDPC-STEM experiments were carried out on a Cs-corrected scanning transmission electron microscope (FEI Titan Cubed Themis G2 300) equipped with a four-quadrant detector and operated at an accelerating voltage of 300 kV. The convergence semi-angle was 15 mrad, the beam current was kept below 0.5 pA, the collection angle ranged from 4 to 22 mrad, and the probe dwell time was 32–64 $\mu$s.

Sheet-like ZSM-5 single crystals were ultrasonically dispersed in pure pyridine or thiophene for 1–2 h, then deposited onto microgrids and heated at 130 °C for 2 h to remove physically adsorbed molecules, leaving only those strongly interacting with the acid sites. In view of the intrinsic stochastic nature of single-molecule adsorption and dynamics, the present work mainly focuses, in the quantitative analysis, on assessing the robustness of the framework reconstruction pipeline. To this end, we selected image regions with little or no visibly adsorbed small molecules, cropped these relatively "clean" framework areas, and used them to evaluate the reconstruction performance of the ZSM-5 zeolite framework.

**AutoMat reconstruction on real ZSM-5 images.** We then fed these relatively clean ZSM-5 framework regions directly into *AutoMat* for end-to-end reconstruction. The results (Fig. 6) show that *AutoMat* can effectively recover the ZSM-5 surface framework from real iDPC-STEM images. On the one hand, the reconstructed CIFs exhibit continuous and well-defined ten-membered-ring channels and the expected framework topology, in close qualitative agreement with the canonical DFT ZSM-5 structure. On the other hand, after aligning the reconstructed structure with the DFT reference, the deviation of framework atomic positions is within approximately 0.2 Å, indicating that *AutoMat* maintains good structural accuracy even under realistic experimental noise and contrast conditions.

Although this test system differs from our main task (2D monolayers) in both material class and imaging conditions, the ZSM-5 experiment demonstrates that *AutoMat* can already exhibit a certain degree of generalization to real experimental data without any task-specific fine-tuning, providing preliminary evidence for its potential deployment on a broader range of real materials systems.

### A.11 Limitations and Future Directions

While AutoMat demonstrates the feasibility of an agentic pipeline for microscopy-to-structure reconstruction, our present scope is intentionally restricted to **2D monolayer systems imaged with**

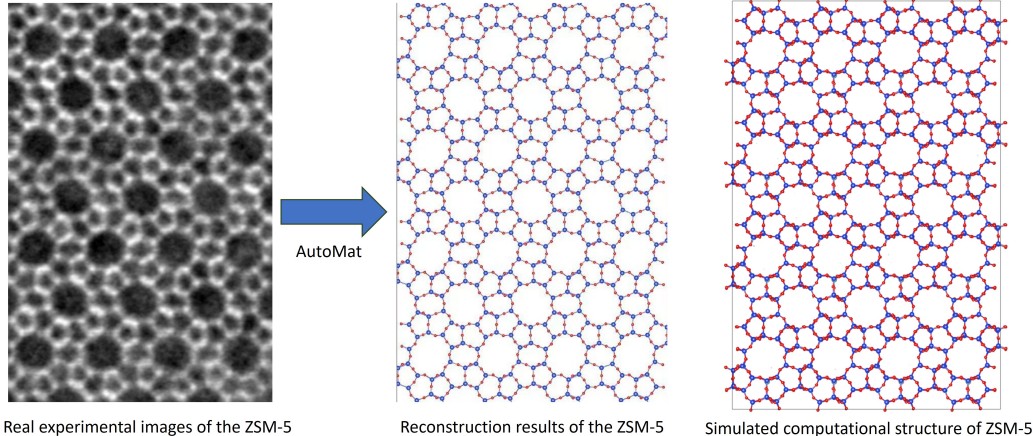

Real experimental images of the ZSM-5     Reconstruction results of the ZSM-5     Simulated computational structure of ZSM-5

Figure 6: **AutoMat reconstruction on real iDPC-STEM images of ZSM-5 zeolite.** (Left) Real experimental iDPC-STEM image of the ZSM-5 framework, showing the characteristic ten-membered-ring pore arrangement. (Middle) Atomic structure reconstructed by *AutoMat* directly from the real STEM image, where the surface framework and pore topology are largely recovered. (Right) DFT-relaxed reference model of ZSM-5.

**iDPC-STEM**. This design yields a stable and physically interpretable testbed, but also introduces several limitations:

- **2D scope and single-projection depth ambiguity.** The current pipeline assumes a single 2D iDPC-STEM projection of a monolayer, exploits the approximately linear relationship between iDPC contrast and atomic number (Z), and leverages this for a deterministic, physics-guided reconstruction. This setting is well suited to 2D materials, but inherently discards z-depth information and therefore cannot yet be directly applied to 3D bulk crystals or multilayer stacks.

- **Element classification uncertainty.** Even in 2D, element assignment still relies on contrast cues and template priors. When elements have similar Z, or when contrast is degraded by low dose or aberrations, different species may become indistinguishable in STEM images. Such misclassification is a major source of reconstruction error and can propagate to downstream property prediction failures.

- **Template retrieval sensitivity to acquisition conditions.** In the current release, the re-trieval library is generated at a fixed pixel scale and spans only a limited range of imaging conditions; template matching is therefore sensitive to mismatches in pixel scale, defocus, and aberrations, which reduces robustness when acquisition conditions deviate from those assumed in the simulations.

- **Heuristic agent policy instead of a learnable one.** At present, the agent uses state-dependent rules and prompt scheduling to select tools and refine parameters. Given the small, semantically stable tool pool and the fact that evaluation signals (RMSD, C.C., S.S., energy errors) are costly to obtain and delayed, this rule-based design already meets our target metrics while preserving interpretability, traceable rollback, and safe self-correction. However, it may not remain optimal once the tool set and task space become significantly larger and more branching.

**Future directions.** To address these limitations, we plan to: (i) integrate **3D-aware imaging and modeling**, such as multi-tilt STEM tomography, to recover depth information and extend AutoMat toward 3D and multilayer structures; (ii) adopt **contrast-sensitive recognition models** and fuse complementary spectroscopy (e.g., EELS/EDX) to reduce element confusion and improve robust element discrimination; (iii) explore **learnable policy optimization**, including reinforcement-learning–based tool selection, once the tool pool and task complexity have grown to a level where

purely rule-based scheduling is no longer sufficient; and (iv) **expand the template-retrieval space and introduce a lightweight pre-retrieval calibration stage** (e.g., for pixel scale and defocus) to reduce dependence on exact acquisition matching and improve robustness across different instruments and imaging settings.

As the first end-to-end framework in this field, AutoMat establishes a foundation for future work in extending to 3D systems, improving atomic species identification, and ultimately enhancing structural fidelity, retrieval robustness, and predictive reliability.

### A.12    USE OF LLMs

In this work, large language models (LLMs) were used solely for text polishing and grammar checking of the manuscript. They were not involved in research design, implementation, or analysis.

