# OpenReview forum: "AutoMat: Enabling Automated Crystal Structure Reconstruction from Microscopy via Agentic Tool Use"
_ICLR.cc/2026/Conference — Submitted to ICLR 2026_

### Official Review · Reviewer_Nie1 · 2025-10-22

**Soundness:** 3
**Presentation:** 3
**Contribution:** 2
**Rating:** 4
**Confidence:** 4

**Summary:**

This paper presents AutoMat, an agent-based framework that directly converts Scanning Transmission Electron Microscopy (STEM) images into Crystallographic Information Files (CIFs), aiming to bridge the gap toward end-to-end structural analysis. The main contribution lies in the agent framework, which consists of four components: MOE-DIVAESR, Template Matching, STEM2CIF, and MatterSim (a sota tool that is not part of the paper’s original contribution). The idea is reasonable; however, several important issues need to be addressed.

**Strengths:**

The overall storyline is reasonable. The end-to-end structural analysis is decomposed into four key steps, and the agent is capable of orchestrating the tools to implement the entire workflow. The validation is multi-faceted and relatively comprehensive.

**Weaknesses:**

The current dataset is quite limited, containing only 2,143 structures, with 450 used for validation. This significantly affects the generalizability of the proposed workflow. Moreover, AI-assisted characterization is indeed a highly practical strategy; however, relying solely on simulation-based validation poses a high risk under a purely theoretical framework. Therefore, validation using experimental real-world data should be incorporated.

**Questions:**

+ As I emphasized in the limitations, please provide experimental validation, at least the case studies, to demonstrate the framework’s effectiveness on real data.

+ It is unclear whether the structures retrieved from the database are used as ground truth in the validation process. If that is the case, then after Template Matching, the system essentially matches a noisy synthetic STEM image to the already recorded crystal structure, which is considered the “ground truth.” In that scenario, the STEM2CIF step seems unnecessary, as it only introduces additional noise and produces a structure that deviates from the retrieved “ground truth.”
This also raises another concern: if the retrieved structures are directly used as reference, then the composition correctness (C.C.) metric will trivially reach 100%, thus losing its discriminative meaning.

+ The validation using formation energy MAE does not provide additional insight, since MatterSim is relatively robust to small structural mismatches, making the results insensitive to moderate reconstruction errors.

+ The novelty of each component is rather limited. The current agent mainly relies on rule-based and prompt-driven tool scheduling, lacking any learnable policy optimization such as reinforcement learning-based tool selection, which would significantly strengthen the framework’s adaptability.

---

> ### Author Response · Authors · 2025-11-13
>
> ### **Q1&W1: Lack analysis of real experimental micrographs and real-image evaluation**
> We agree with the reviewer and outline the constraints and our mitigation:
> * **Public data availability & compliance**: Many public iDPC-STEM images for 2D materials are inaccessible (broken links) or only available as compressed, annotated figures with pixel distortion; direct reuse also raises copyright/ethics concerns.
> * **Practical difficulty of new acquisitions**: High-quality, dose-limited, large-field iDPC-STEM requires sample preparation, microscope booking, and repeated acquisitions for calibration/quality—an inherently time-consuming process involving specialized equipment and personnel.
>
> **Planned minimal real-image add-on** (during rebuttal/revision, on a best-effort basis):
> We will **add 1–2 large-field real iDPC-STEM case studies**. We kindly ask for understanding of the practical constraints and will strive to deliver a concise yet informative validation.
>
> ### **Q2: Does AutoMat retain template-free reconstruction capability when template matching is used only as a prior?**
>
> We appreciate the reviewer’s careful observation. We acknowledge this is indeed a key issue in our current pipeline. Concretely, **template matching** works as follows: domain experts first prepare a set of **candidate structures**, project them along specific **zone axes/planes** to produce **projected label images**, and then use these labels to **assist** subsequent reconstruction. Importantly, this step is **not unique to our work**; in practical electron microscopy analysis, researchers commonly curate a **candidate database or structural prior** to avoid large deviations during interpretation.
>
> We also conducted an **ablation** in the appendix A.4 Table 3 where **template matching is completely removed**: the metrics **degrade noticeably**, yet the system still **outperforms** most baselines. This indicates that, even **without templates**, our pipeline retains non-trivial capability to **reconstruct the minimal primitive cell** from denoised images (via STEM2CIF). The reviewer’s additional point is equally important: if we **over-rely on templates**, the manually designed prior can **bias** the prediction away from the true structure. This is precisely one of our **future focuses** for improvement (we will clarify this limitation and direction in the revision without altering the paper’s conclusions).
>
> **Table 3.** Energy per Atom MAE (meV/atom) under different ablation settings.
>
> | **Method**                       | **Tier 1** | **Tier 2** | **Tier 3** |
> | -------------------------------- | ---------: | ---------: | ---------: |
> | *No Denoising (w/o MOE-DIVAESR)* |       6584 |       2616 |        938 |
> | *No Template Matching*           |        617 |        608 |        672 |
> | **Full Pipeline (AutoMat)**      |    **344** |    **320** |    **333** |
>
> ### **Q3: Formation-energy MAE adds little insight, since MatterSim’s robustness to small mismatches makes it insensitive to moderate reconstruction errors.**
> We appreciate the concern that formation-energy MAE may be relatively insensitive to moderate structural deviations due to the robustness of the ML interatomic potential. In our study, energy MAE is intended to measure **end-to-end usability**—whether the reconstructed structure is adequate for downstream property calculation—rather than to single-handedly resolve fine-grained geometric differences. To mitigate the limitations of any single metric, we report **complementary structural metrics** (e.g., projected lattice error *RMSDxy*, composition correctness *C.C.*, and structure success *S.S.*), which directly capture atomic placement, lattice fit, and stoichiometry, thereby **complementing** the energy MAE.
>
> Moreover, our upper-bound comparison (ground-truth CIF + MLIP) and tiered analyses indicate that the gap to the upper bound is primarily driven by **reconstruction error**, not by MLIP “masking” effects. To further address the reviewer’s point, we will **emphasize and extend** more geometry-sensitive diagnostics (e.g., pre- vs. post-relaxation energy drop *ΔE*, force residuals) and provide correlation analyses with structural errors, making the relationship between energy- and structure-side metrics more transparent and complete.

---

> ### Author Response · Authors · 2025-11-13
>
> ### **Q4: Components show limited novelty; the agent relies on rule/prompt-based tool scheduling and lacks learnable policy optimization (e.g., RL tool selection), which limits adaptability.**
> #### **1. Core Motivation and Contribution**
> We would like to begin by clarifying the core motivation of this paper. Scanning Transmission Electron Microscopy(STEM) offers sub-atomic resolution, a domain expert needs **6-8 hours** to analyze a single image, significantly slowing scientific discovery. Machine learning has been widely adopted in analyzing STM to compression this operation into minutes. However, current ML enpowered automated microscopy systems are fragmented, treating acquisition, reconstruction, and analysis as isolated steps. It is non-trivial to close the "acquisition-analysis-feedback" loop. LLM agent is a perfect candidate to do this job, yet there is currently no work trying to bridge the gap.
>
> We release the first agentic system that **(1)** offers a standardised toolchain, testbed, and benchmark for STEM "acquisition-analysis-feedback" task, and **(2)** systematically evaluates the capability of agents to close the loop.
>
> Our core innovation is the **first agentic framework** that autonomously invokes a toolchain for STEM image-to-structure conversion and property prediction in a closed loop. Below shows each module in our agent framework:
> | **Method/Model**   | **Description**                                                     | **Source/Reference**                                  |
> |--------------------|---------------------------------------------------------------------|-------------------------------------------------------|
> | MOE-DIVAESR        | Pattern-based denoising model                     | New proposed                                |
> | Template Matching  | A conventional module for structural priors.                        | Common method in theoretical texts                    |
> | STEM2CIF           | Structure and primitive cell reconstruction. | New proposed                                  |
> | MatterSim          | Structural optimization and property prediction. | Microsoft Research team                  |
>
> We firmly believe the above contribution is well-suited for ICLR@AI4S track.
>
> #### **2. When to adopt “learnable policy optimization” — forward plan**
> We appreciate the suggestion. We do not introduce RL-based tool selection at this stage because the **action space is small**, tool semantics are **stable**, and the **evaluation signals** (RMSD, C.C., S.S., energy errors) are **costly to obtain and delayed**. With a **limited tool pool** and **well-defined decision rules**, **state-dependent rule/prompt scheduling** already meets the target metrics while preserving **interpretability and safety** (e.g., traceable rollback and parameter self-correction). In short, at the current scale the **added complexity and training cost** of RL would not yield **proportionate gains**.
> That said, we agree that **learnable policies** become valuable when the **tool pool expands substantially**, tasks exhibit **branching explosion**, or involve **long-range cross-modal dependencies**. In those settings, incorporating **reinforcement learning** for tool selection and policy optimization would be **appropriate and beneficial**, and we will make this upgrade path explicit in the **discussion/future work** section.

---

> ### Author Response · Authors · 2025-11-19
>
> ### **Addition of real-image validation**
> We fully agree with the reviewer that AutoMat should be validated on real experimental data. However, under our current constraints in sample preparation and microscope time, it is experimentally very difficult—within this revision period—to prepare new 2D monolayer samples from scratch and acquire high-quality, stable iDPC-STEM images.
>
> Given these practical constraints, we still wished to provide a real experimental test, and therefore chose the ZSM-5 zeolite, whose crystallographic model is well established, as a compromise. In the revised manuscript, Section 5.2 “Real iDPC-STEM Case Study and Human-Expert Effort” and Appendix A.10 show that, on ZSM-5 iDPC-STEM images acquired on the same Cs-corrected STEM under dose-limited conditions, AutoMat can perform denoising and structure reconstruction without manual parameter tuning, and that the projected lattice and channel framework of the resulting CIF are qualitatively consistent with the known ZSM-5 structure.
>
> ---
>
> ### **Refinement of limitations and future directions**
> In the revised manuscript, we have already clarified this point as part of our future plans in Appendix A.11 “Limitations and Future Directions”: when the tool pool expands substantially, task flows exhibit large branching factors, or long-range cross-modal dependencies arise, introducing learnable policies will become increasingly important and valuable, and we regard this as an important direction for future extension.
>
> We once again thank the reviewer for these valuable suggestions; the newly added real-image experiments and the expanded discussion of “limitations and future work” in the revised version directly benefit from these comments.

---

> > ### Comment · Reviewer_Nie1 · 2025-11-23
> > **Further Comment**
> >
> > I appreciate the authors’ efforts. My main concern in `Q2` is about the definition of the ground truth. Is the structure recorded in the database used as the ground truth for the final validation?
> >
> >
> > In Step 2, Image Template Matching retrieves several candidates from the ground-truth pool. The subsequent STEM2CIF conversion and relaxation introduce perturbations to the original ground-truth structures, producing an approximate structure that no longer strictly corresponds to any entry in the crystal-structure database. This makes the final validation appear more like a process of reconstructing the structure rather than evaluating a direct match or mismatch.

---

> > > ### Author Response · Authors · 2025-11-23
> > > **Response: Clarification on Ground Truth Definitions and Robustness in the Absence of Templates**
> > >
> > > **Response: Clarification on Ground Truth Definitions and Robustness in the Absence of Templates**
> > >
> > > We sincerely thank the reviewer for raising this important and insightful question. Your understanding of our methodology is entirely correct. We provide a point-by-point clarification below:
> > >
> > > **(1) On whether the database structure serves as the Ground Truth**
> > >
> > > Yes, the structures recorded in the databases (C2DB / Materials Project / OpenCrystal) serve as the **Ground Truth** for our final structural evaluation.
> > > * **Clarification on Matching**: It is important to note that the verification process does not rely on direct CIF-to-CIF file matching. Instead, matching is performed via the projected images along the **[001] direction**.
> > > * While the template library contains a vast number of candidates (naturally including the actual ground truth structure), this approach aligns with the traditional workflow in electron microscopy structural analysis: screening and matching the candidate template that best fits the observed imaging data from a pool of possibilities.
> > >
> > > **(2) On the "Perturbations" introduced by STEM2CIF and Structural Relaxation**
> > >
> > > The reviewer is correct that the final structure exhibits slight deviations from the original database entries following the STEM2CIF conversion and structural relaxation. This design is **intentional** and serves specific purposes:
> > >
> > > * **Physical Realism**: 2D materials are not strictly planar; they often exhibit subtle variations (such as buckling or corrugation) along the **z-axis**.
> > > * **Optimization**: Moderate initial perturbations facilitate the subsequent relaxation steps, allowing the system to adjust atomic positions toward a more reasonable local geometry based on image features.
> > > * **Robustness**: This approach helps the model accommodate noise, projection distortions, and the loss of z-axis depth information inherent in real STEM images.
> > >
> > > Therefore, the final output should be interpreted as a **reconstructed estimation** based on the input image, rather than a verbatim copy of the database entry. This aligns perfectly with the paper's objective of evaluating "structural reconstruction accuracy."
> > >
> > > **(3) Is AutoMat still effective if the Ground Truth is absent from the database?**
> > >
> > > We fully appreciate the reviewer's concern: Can the agent function correctly without a ground truth template?
> > >
> > > To address this, we conducted a dedicated **Ablation Study** (see **Appendix A.4**):
> > > * **Setup**: We **completely removed the Template Retrieval module**.
> > > * **Result**: The results demonstrate that AutoMat retains significant structural reconstruction capabilities even without template assistance. This proves that the system is not merely relying on a "lookup table" approach but possesses intrinsic reconstruction capabilities.
> > >
> > > Furthermore, in the newly added **Real-world iDPC-STEM Validation** (see **Appendix A.10**), we performed reconstruction directly from experimental images **without using template matching**. The system successfully derived reasonable CIF structures (as shown in **Figure 6 of Appendix A.10**).
> > >
> > > Collectively, these results confirm that AutoMat remains effective and robust even when exact ground truth templates are unavailable.

---

> > > > ### Comment · Reviewer_Nie1 · 2025-11-23
> > > > **reply**
> > > >
> > > > I sincerely thank the authors for the additional clarification. After consideration, I will maintain my initial assessment, which remains slightly below the acceptance threshold in my view.
> > > >
> > > > Nevertheless, I truly appreciate the authors’ efforts to improve the manuscript, and I would not oppose its acceptance if the AC finds it suitable.

---

> > > > > ### Author Response · Authors · 2025-11-23
> > > > >
> > > > > We sincerely thank the reviewer for the follow-up and for acknowledging our efforts during the revision process. We also appreciate your openness regarding the potential acceptance of our paper.
> > > > >
> > > > > While we respect your current position, we wish to emphasize that the newly added experiments—specifically the **ablation study on template-free performance** and the **real-world iDPC-STEM validation**—were conducted to directly address the core concerns regarding robustness and real-world applicability. Consequently, we believe the manuscript has been substantially strengthened compared to the initial submission.
> > > > >
> > > > > We respectfully hope that this solid evidence might warrant a reconsideration of the rating, or a stronger recommendation to the AC, as we believe the primary limitations previously identified have been effectively addressed. However, should you still feel the work falls short of the acceptance threshold, we remain genuinely grateful for your constructive suggestions, which will undoubtedly help refine our research endeavors going forward.

---

> > > > > > ### Comment · Reviewer_Nie1 · 2025-11-25
> > > > > >
> > > > > > Thank you for the summary. I am unable to provide a more positive evaluation.
> > > > > >
> > > > > > I still have concerns regarding the practical application scenario. First, the limited experimental validation raises questions about the reliability and real-world applicability of the method. I fully understand that it may be challenging to prepare a comprehensive experimental setup during the revision period; therefore, from my perspective, I can only provide a conservative assessment.
> > > > > >
> > > > > > Another concern relates to the ground-truth retrieval and denoising workflow. Although the ablation study indicates that the absence of structural coverage in the database has limited impact on AutoMat’s performance, the inference process still fundamentally depends on structure retrieval rather than explicit structure reconstruction. This reliance suggests that downstream applications may be constrained by the completeness of the database.
> > > > > >
> > > > > > I recommend considering whether, in future work, you could provide a usable platform that allows STEM experts to apply the method directly. Such a tool would create new opportunities for real-world evaluation and would strengthen confidence in the method's practical impact, especially given its stated focus on applied outcomes.

---

> > > > > > > ### Author Response · Authors · 2025-11-25
> > > > > > >
> > > > > > > We sincerely thank the reviewer for the careful, candid, and constructive evaluation of our work. We respond to your concerns point by point as follows.
> > > > > > >
> > > > > > > ---
> > > > > > >
> > > > > > > ### **(1) On the difficulty and practical cost of experimental validation**
> > > > > > >
> > > > > > > We fully understand the reviewer’s expectation for broader real-data validation across multiple material systems, especially systematic experiments on well-characterized 2D monolayers with known crystallographic information. However, in practice, securing beamtime on a Cs-corrected STEM is associated with substantial time and financial cost.
> > > > > > >
> > > > > > > At our facility, the microscope fee is **approximately 700 USD/hour**.
> > > > > > >
> > > > > > > For the ZSM-5 real experiment included in this revision, in order to obtain a batch of samples suitable for iDPC-STEM imaging, we needed:
> > > > > > >
> > > > > > > * **Around 3 days** of accelerated material synthesis and sample preparation;
> > > > > > > * **About 6 consecutive hours of microscope time** to acquire a set of iDPC-STEM images with sufficient resolution and stability.
> > > > > > >
> > > > > > > Even for a single material system, this already represents a considerable investment of time and cost. Therefore, we kindly ask for the reviewer’s understanding that, within this revision cycle, it is very difficult for us to extend to more material systems. Under these practical constraints, we still made an effort to include a representative real microscopy case in order to strengthen the empirical basis of AutoMat.
> > > > > > >
> > > > > > > ---
> > > > > > >
> > > > > > > ### **(2) On ground-truth retrieval and dependence on the structural database**
> > > > > > >
> > > > > > > We fully appreciate the reviewer’s concern about “ground-truth retrieval” and the completeness of the database: the current workflow indeed contains a template-retrieval step based on a structure library, and in real applications the coverage of that library will have an impact.
> > > > > > >
> > > > > > > At the same time, we would like to clarify that AutoMat is **not** a simple “lookup system”. As we explained in our response to Reviewer NmQd (Q3 & Q4), the core of AutoMat follows a **“threshold gating + agentic re-planning”** scheme, rather than a fixed “retrieve first and stop” script:
> > > > > > >
> > > > > > > > *“Threshold gating first, LLM re-planning second.… If repeated attempts still fail within a capped budget, the agent falls back to template retrieval to secure a robust prior.”*
> > > > > > >
> > > > > > > > *“Why an agent instead of a pure script? … our agent alters strategy based on textual feedback, going beyond deterministic branching.”*
> > > > > > >
> > > > > > > In other words, **template retrieval only serves as a fallback prior after multiple rounds of direct reconstruction have failed**, while the main path remains an explicit reconstruction process driven by tool feedback and agent re-planning. In the revised manuscript (Appendix A.11), we explicitly list issues such as “database coverage” and “robustness to parameter mismatch” as current limitations and as key directions for future improvement (e.g., expanding structural coverage, and improving robustness to defocus/aberration/pixel-scale variations).
> > > > > > >
> > > > > > > ---
> > > > > > >
> > > > > > > ### **(3) On providing a usable platform for microscopy experts**
> > > > > > >
> > > > > > > We strongly agree with the reviewer’s suggestion about offering a platform that can be directly used by STEM experts. In line with this vision, **our code and the STEM2Mat benchmark will be fully open-sourced and regularly maintained**. We believe that:
> > > > > > >
> > > > > > > * Only with openness can more electron-microscopy experts and theory researchers conveniently apply AutoMat to their own data and tasks;
> > > > > > > * And only in this way can we truly promote broad, real-world validation and collaborative improvement in the AI4S community.
> > > > > > >
> > > > > > > Once again, we sincerely thank the reviewer for the valuable comments and for continuing the discussion. We hope that our explanation can help to alleviate, at least to some extent, your concerns and doubts.

---

> > > > > > > > ### Comment · Reviewer_Nie1 · 2025-11-25
> > > > > > > >
> > > > > > > > I appreciate the authors’ efforts to further refine this work.
> > > > > > > >
> > > > > > > > I acknowledge your claims, and regarding the platform, **I am pleased to see that the authors have the ambition to move toward real-world applications**. However, **open-sourcing the code alone may not be sufficient**. If possible, I encourage you to consider releasing a user-friendly interface, perhaps based on free or lightweight computational backend services, to make the tool more accessible for broader use.
> > > > > > > >
> > > > > > > > Based on the revisions and improvements, I am willing to raise my score to `6`, supporting the paper for further development.

---

> > > > > > > > > ### Author Response · Authors · 2025-11-25
> > > > > > > > >
> > > > > > > > > We sincerely thank the reviewer for the thoughtful follow-up assessment and for the constructive vision regarding real-world applicability. We truly appreciate your scientific rigor and your willingness to engage deeply with our work throughout the review process. Your suggestions—especially the encouragement to move beyond open-source code and toward a more user-friendly, accessible interface—are both insightful and inspiring, and they will strongly influence our next-stage development.
> > > > > > > > >
> > > > > > > > > We are grateful for your careful evaluation, and we genuinely appreciate your decision to raise the score. Your feedback has been highly valuable to us, and the discussion has meaningfully shaped the future direction of AutoMat.

---

### Official Review · Reviewer_rSdp · 2025-10-31

**Soundness:** 2
**Presentation:** 2
**Contribution:** 2
**Rating:** 4
**Confidence:** 3

**Summary:**

The work presented proposes AutoMat, an LLM agentic pipeline that converts a single STEM image into a simulation‑ready crystal structure (CIF) and then predicts formation energy via an ML interatomic potential. The agent coordinates four tools: (1) MOE‑DIVAESR for pattern‑adaptive denoising, (2) template retrieval from a library of simulated STEM projections, (3) STEM2CIF for symmetry‑aware reconstruction, and (4) MatterSim for energy relaxation/prediction.

**Strengths:**

* The image → structure → property loop is valuable and timely for AI-for-science, integrating denoising, retrieval, symmetry-aware reconstruction, and MLIP relaxation in a single agent is a meaningful system contribution.

* The presented benchmark shows that AutoMat outperforms domain toolkits and VLMs on structural accuracy

**Weaknesses:**

* Internal inconsistency with dataset and tiering count, its stated that there is  450-case test split, yet in section 3.3 its reported that tier sizes 35, 456, 79.
* The pipeline is motivated by experimental workflows, yet all quantitative results use abTEM simulations (fixed pixel size, dose ranges, aberration/noise models). No results are shown on real STEM micrographs
* AtomAI is a segmentation/localization library, not a CIF reconstructor; holding it to CIF-level metrics seems mismatched.
* The upper bound using the ground-truth CIF and MLIP shows an MAE of 48–57 meV/atom, substantial residual versus DFT even for perfect structures. Reported AutoMat MAEs (~320–330 meV/atom) thus reflect a mix of (i) structural error, (ii) MLIP bias vs. DFT, and (iii) sensitivity to starting geometries.
* The appendix reports agent/backbone swaps with ±3% variation and the retry mechanism, but there is no controlled ablation showing the marginal benefit of MOE-DIVAESR vs. a strong single denoiser, nor of symmetry constraints, nor of retrieval vs. no-retrieval. Given the four modules, their individual contributions should be quantified.

**Questions:**

* Considering the 450 test cases vs. 35/456/79 tier sizes (sum 570). Which is correct, and what numbers were used in Tables 1–2? If 570 is correct, where do the extra cases come from?
* How does AutoMat perform if the ground-truth structure is not in the retrieval library? Can you report structural metrics and energy MAE in that setting?
* Can you add real-image evaluation on well-characterized monolayers with known CIFs?
* Is it possible to provide module-wise ablations?
* How sensitive is retrieval to microscope parameter mismatch (defocus, aberrations, pixel scale)? Any simple calibration step before retrieval?

---

> ### Author Response · Authors · 2025-11-13
>
> We thank Reviewer rSdp for the constructive and detailed comments.
>
> ### **Q1&W1: Clarification on “450 vs. 35/456/79” and detailed tiering rules**
>
> Thank you for the careful reading. **Bottom line:** our test split contains **459 unique structures (unique CIF cases)** (the “450+” in the manuscript was an approximation). By contrast, the **35/456/79** figures in Sec. 3.3 are **tier-wise counts of samples under different tier conditions**, and **tiers overlap in terms of unique structures**. The apparent “extra 111” arises from a different counting granularity: these are **not additional structures**, but **distinct image instances of the same structure acquired/simulated at different electron doses**. Results in Tables 1–2 are computed **per simulated characterization image instance** within each tier; any **overall/Avg** figure aggregates **over all instances**.
>
> More specifically, we define a **case** as a **unique CIF** (fixed composition, lattice, and atomic coordinates). An **instance** is **one iDPC-STEM image of a given case** under a **specific microscope setting** (e.g., dose), paired with its ground truth. Because a single case can be simulated/acquired **at multiple doses or settings**, **one case may correspond to multiple instances**, which explains why the summed tier counts differ from the number of unique structures.
>
> Crucially, **tiers are defined over instances, not structures**: we stratify **by composition and imaging dose** (e.g., unary + high dose → Tier 1; binary or moderate dose → Tier 2; ternary and low dose → Tier 3). Consequently, **different instances of the same structure** produced at different doses can fall into **different tiers**, which is why the **sum across tiers exceeds the number of unique cases**. In addition, the **train/validation sets are strictly disjoint from the test set at the case level**; no test case (nor its instances) appears in train/val, preventing leakage. To avoid confusion, we will clarify the **case vs. instance** distinction in the revised manuscript (main text, table notes, and footnotes).
>
> ### **Q3&W2: Additional analysis of real experimental micrographs and real-image evaluation**
> We agree with the reviewer and outline the constraints and our mitigation:
> * **Public data availability & compliance**: Many public iDPC-STEM images for 2D materials are inaccessible (broken links) or only available as compressed, annotated figures with pixel distortion; direct reuse also raises copyright/ethics concerns.
> * **Practical difficulty of new acquisitions**: High-quality, dose-limited, large-field iDPC-STEM requires sample preparation, microscope booking, and repeated acquisitions for calibration/quality—an inherently time-consuming process involving specialized equipment and personnel.
>
> **Planned minimal real-image add-on** (during rebuttal/revision, on a best-effort basis):
> We will **add 1–2 large-field real iDPC-STEM case studies**. We kindly ask for understanding of the practical constraints and will strive to deliver a concise yet informative validation.
>
> ### **W3: Clarification on the Scope and Evaluation Metric for AtomAI**
> In this paper, we do **not** compare AtomAI against the agentic system on **all** metrics because AtomAI does **not** cover the full end-to-end scope considered here. AtomAI is positioned as a **module-level** segmentation/localization toolkit. Accordingly, we **convert its segmentation and atomic-localization outputs** and use them as a **step/component-level baseline** (e.g., for **localization/atomic-peak detection**), rather than as a direct comparator to an end-to-end agentic system. We evaluate using the **RMSD on the CIF’s (xy) projection**, which **succinctly captures** the accuracy of both **atom-type assignment** and **positional localization**.

---

> ### Author Response · Authors · 2025-11-13
>
> ### **Q2 & W5 & Q4: Performance when the ground-truth structure is absent from the retrieval library; reporting structural metrics and energy MAE; and module-wise ablations**
>
> We provide **module-wise ablations** in **Appendix A.4**. Addressing the reviewer’s key concern—**removing template retrieval**—**Table 3** shows that disabling this module **degrades** end-to-end performance and **substantially increases** the formation-energy **MAE**, yet the system **still outperforms** most prevailing baselines. For **genuinely novel structures** that are **not** in the database, the pipeline **relies entirely on the denoised STEM image**, and **STEM2CIF** reconstructs the structure. We **support automatically recovering and rebuilding the minimal primitive cell from a supercell**, enabling **template-free** lattice and composition estimation. The relevant numbers are given in the **“*No Template Matching*”** row of the ablation.
>
> **Table 3.** Energy per Atom MAE (meV/atom) under different ablation settings.
>
> | **Method**                       | **Tier 1** | **Tier 2** | **Tier 3** |
> | -------------------------------- | ---------: | ---------: | ---------: |
> | *No Denoising (w/o MOE-DIVAESR)* |       6584 |       2616 |        938 |
> | *No Template Matching*           |        617 |        608 |        672 |
> | **Full Pipeline (AutoMat)**      |    **344** |    **320** |    **333** |
>
> ### **Q5: Sensitivity of retrieval to microscope parameter mismatch (defocus, aberrations, pixel scale), and whether a simple pre-retrieval calibration exists**
>
> We appreciate the reviewer’s constructive question. In the current release, the retrieval library is generated at a **fixed pixel scale**, and we **vary electron dose** to cover different SNR regimes. Consequently, template retrieval is **sensitive** to **pixel-scale, defocus, and aberration** mismatches: these factors are **not** systematically modeled in this round of simulations, and deviations outside the preset range reduce robustness and comparability.
>
> Regarding a “simple pre-retrieval calibration”: **we do not include an additional automatic calibration step at present**. Our experiments assume that the input images’ **resolution (pixel scale) and acquisition conditions** are **closely matched** to those used for the retrieval library. This also explains why, for **real-image evaluation**, we rely on **experienced operators** to acquire data under conditions that align with our current assumptions.
>
> We will **make this limitation explicit** in the revision and list “expanding the retrieval space to include defocus/aberrations/pixel scale and adding a pre-retrieval calibration stage” as future work to reduce dependence on acquisition matching and improve robustness across instruments and settings.

---

> ### Author Response · Authors · 2025-11-19
>
> ### **Addition of real-image validation**
> We fully agree with the reviewer that a system intended for closed-loop experimental microscopy should, in an ideal setting, be validated on multiple real material systems, especially through systematic experiments on 2D monolayers with known CIFs. However, under our current constraints in sample preparation and microscope time, it is experimentally very challenging to prepare new 2D monolayer samples from scratch and acquire high-quality, stable iDPC-STEM images within this revision cycle.
>
> Given these practical limitations, we still wanted to provide a real experimental test. We therefore chose the ZSM-5 zeolite—whose crystallographic model is well established—as a compromise. In the revised manuscript, Section 5.2 “Real iDPC-STEM Case Study and Human-Expert Effort” and Appendix A.10 present results showing that, on dose-limited ZSM-5 iDPC-STEM images acquired on the same Cs-corrected STEM, AutoMat can perform denoising and structure reconstruction without manual parameter tuning. The projected lattice and pore framework of the resulting CIFs are qualitatively consistent with the known ZSM-5 structure. We explicitly state in the manuscript that this experiment is only an initial feasibility check of AutoMat on real data, and not a substitute for systematic validation across multiple real 2D monolayer systems. We also highlight “conducting more systematic real-image validation on multiple monolayer materials with known CIFs” as one of the key priorities for future work.
>
> ### **Refinement of limitations and future directions**
> In the revised manuscript, we have made this point explicit as a limitation in Appendix A.11 “Limitations and Future Directions,” and we discuss it as an important avenue for future extension. Specifically, we plan to expand the template library to cover a more realistic range of defocus and aberration parameters, introduce simple calibration steps for pixel scale and imaging conditions, and explore more robust representation-learning methods to reduce the impact of moderate parameter drift on retrieval. We believe these efforts will substantially improve the practical stability of AutoMat in real experimental settings.
>
> We once again thank the reviewer for these important suggestions; the newly added real-image experiments and the expanded discussion on “limitations and future work” in the revised manuscript directly benefit from these valuable comments.

---

> ### Author Response · Authors · 2025-11-27
>
> Dear Reviewer rSdp,
>
> I hope this message finds you well.
>
> As the discussion period is nearing its end with less than seven days remaining, I wanted to ensure we have addressed all your concerns satisfactorily. If there are any additional points or feedback you would like us to consider, please let us know. Your insights are invaluable to us, and we are eager to address any remaining issues to improve our work.
>
> I also realize this might be a holiday period for you. If you are celebrating Thanksgiving, I apologize for the interruption and wish you a wonderful holiday. We appreciate your time and effort in reviewing our paper given the busy schedule.
>
> Best regards

---

> ### Author Response · Authors · 2025-11-28
>
> Dear Reviewer rSdp,
>
> Thank you once again for taking the time to review our work and for providing valuable feedback. We understand that, due to certain constraints, the system may no longer allow score updates at this stage of the process. Nevertheless, we genuinely value your comments, and before the final response deadline, we hope to ensure that any remaining questions or concerns you may have are fully addressed.
>
> If there are **any unresolved issues, points requiring further clarification, or aspects that you believe could benefit from additional explanation**, we would be more than willing to provide further responses during this final window—even if these can no longer affect the score. Your insights are highly valuable to us, both for this submission and for the future development of AutoMat.
>
> We sincerely appreciate your time, expertise, and the constructive feedback you have provided.
>
> With kind regards

---

### Official Review · Reviewer_WLyJ · 2025-11-02

**Soundness:** 2
**Presentation:** 2
**Contribution:** 1
**Rating:** 2
**Confidence:** 4

**Summary:**

The paper introduces AutoMat, an agent-based system that claims to reconstruct atomic crystal structures directly from Scanning Transmission Electron Microscopy (STEM) images and predict material properties. The proposed workflow integrates, denoising, a template-matching retrieval module, STEM2CIF for lattice reconstruction, and MatterSim for energy prediction—managed by an LLM “agent” for orchestration. Additionally, the authors present STEM2Mat-Bench, a synthetic benchmark dataset of 450 annotated samples, used to evaluate reconstruction accuracy and formation-energy errors. Results show AutoMat outperforming other baselines methods in structural RMSD and energy MAE metrics.

**Strengths:**

The problem from STEM images to downstream atom position and properties is very useful in connecting Experiments with simulation's.

**Weaknesses:**

a) Overstated novelty. Many ideas(agenticEM2CIF, agentic orchestration) are incremental extensions of existing frameworks such as SciLink, MicroscopyGPT and Mdcrow.

b) Benchmark validity. The comparison between agentic and non-agentic systems is not apples-to-apples. LLM-VLM baselines are not designed for such end-to-end tasks, making the reported “order of magnitude” improvement misleading. Also AtomAI is like python module with Neural nets like “unet” to segment images, how is it fair to compare it with agentic systems? Rather a study over different llm's orchestrating the same "tools" would be more useful. Basically answering questions like what models perform better or worse, do domain models beat bigger non domain models?

c) Lack of experimental(data collected on Microscope) validation, severely limits scientific impact.

 d) Unclear originality of released data. Since the dataset is built on existing repositories (C2DB, OpenCrystal), the “data release” claim is more of a repackaging effort.




references :
- AtomAI - https://github.com/pycroscopy/atomai,
- SciLink - https://github.com/ziatdinovmax/SciLink
- Mdcrow - https://github.com/ur-whitelab/MDCrow
- MicroscopyGPT - notebook present at  : https://github.com/atomgptlab/atomgpt

**Questions:**

Please see weaknesses.

---

> ### Author Response · Authors · 2025-11-13
>
> We thank Reviewer WLyJ for the thoughtful feedback.
>
> ### **W1: On the claim that novelty is overstated / merely incremental**
> We conducted a thorough survey of tools and models applicable to STEM analysis (e.g.,AtomAI, MicroscopyGPT, chemagent). Existing systems either remain at **text-level QA/description** (e.g., MicroscopyGPT still outputs text that requires **manual** conversion to CIF) or do not realize a **closed-loop automation** from **image → structure (simulation-ready CIF) → property**. This gap motivates our work: combining domain-specific vision/physics modules with agent orchestration to produce **standardized CIFs** ready for computation and complete property prediction—beyond generating text.
>
> #### **Novelty and Contribution**
>
> We begin with the motivation. While STEM offers sub-atomic resolution, a domain expert typically spends **6–8 hours** to analyze a single image, which slows discovery. Although ML has been adopted for microscopy, current systems are **fragmented**—acquisition, reconstruction, and analysis are isolated, making the **acquisition–analysis–feedback** loop non-trivial. LLM agents are natural candidates to close this loop, yet, to our knowledge, no prior work bridges it end-to-end.
>
> We release the first agentic system that **(1)** provides a standardized toolchain, testbed, and benchmark for the STEM **acquisition–analysis–feedback** task, and **(2)** systematically evaluates LLM-based agents’ capability to close the loop.
>
> Our core contribution is the **first agentic framework** that autonomously invokes a toolchain to convert STEM images to structures and predict properties in a closed loop. The modules are:
>
> | **Method/Model**  | **Description**                                    | **Source/Reference**    |
> | ----------------- | -------------------------------------------------- | ----------------------- |
> | MOE-DIVAESR       | Pattern-based denoising model                      | New proposed            |
> | Template Matching | Conventional module for structural priors          | Common method           |
> | STEM2CIF          | Structure and primitive-cell reconstruction to CIF | New proposed            |
> | MatterSim         | Structural optimization and property prediction    | Microsoft Research team |
>
> We believe these contributions fit the ICLR@AI4S track: a **reusable toolchain and benchmark** together with an **end-to-end, closed-loop** from image to simulation-ready CIFs and properties—going beyond systems that only **produce text**.
>
> ### **W2: On the validity of the benchmark: Are the comparisons apples-to-apples? VLM/LLM baselines are not designed for end-to-end pipelines; is a direct comparison with AtomAI fair? Would it be more meaningful to compare different LLM orchestrators under the same toolchain?**
>
> #### **1. Clarification on VLM Baseline Comparison**
> Our comparison with zero-shot VLMs was not a performance contest, but a test of a fundamental hypothesis: **Can current general-purpose multimodal models solve this complex scientific task purely through intrinsic reasoning, without any tool assistance?**
>
> To isolate “scientific reasoning” from information lookup, we carefully controlled and guided the setup:
>
> * We **supplied rich priors** (e.g., elemental composition and material type) and **prompted the model to follow a scientific reasoning path**;
> * We **disabled external search**, preventing answers obtained via retrieval.
>
> Moreover, to assess whether **fine-tuning a multimodal model on existing data** would suffice, we evaluated **MicroscopyGPT** and—**as required by that model**—provided not only the elemental species but also **coordination ratio information**. The results indicate that **limitations persist even under these conditions**, and the model still struggles to reliably complete the full loop from **image → structure (CIF) → property**. This observation motivates our **agent-based framework**, which leverages **specialized tools plus orchestration** to address the gaps of general-purpose models in complex scientific workflows.
>
>
> #### **2. On the “Fairness” of Comparing AtomAI**
>
> In our paper, we **do not** compare AtomAI against the agent system on **all** metrics, because AtomAI **does not cover** the full end-to-end scope considered here. AtomAI is positioned as a **module-level** segmentation/localization toolkit. Accordingly, we treat it as a **step-/component-level baseline** (e.g., for **localization/atomic-peak detection**) rather than a direct comparator to an end-to-end **agentic** system.
>
> To prevent misinterpretation, we will **clarify its role and scope** in the manuscript:
>
> * We report comparisons **only within the functions AtomAI is designed for**, using appropriate **local metrics**;
> * We **do not** extrapolate “**module vs. system**” results into claims of **system-level superiority**.

---

> ### Author Response · Authors · 2025-11-13
>
> #### **3. Overall Sensitivity to the Choice of LLM/VLM Agent**
> To address the reviewer’s suggestion of comparing **different LLM/VLM orchestrators under the same toolchain**, we conducted such comparisons under **identical tools and evaluation settings**. Due to space limits, the results are reported in **Appendix Tables 5, 6, and 7**. The main takeaways are:
>
> 1. **Core scientific metrics are very similar.**
>
>    * **Table 5 (Energy MAE)**: the average MAE across controllers falls in a **narrow range** (approximately *308–314 meV/atom*).
>    * **Table 6 (Structural metrics)**: RMSDxy, composition correctness (C.C.), and structure success (S.S.) are **largely aligned** across tiers.
>      This indicates that, **given the same toolchain**, the choice of LLM/VLM as the controller has **limited impact** on final scientific quality.
> 2. **Differences appear mainly in orchestration behavior, not final quality.**
>
>    * **Table 7 (invocations/success/retries)** shows notable variations in **calling patterns, success rates, and retry counts**, reflecting differences in **scheduling and fault tolerance** (e.g., some controllers achieve higher success with fewer retries).
>    * These behavioral differences aid interpretability and robustness analysis, but **do not translate** into substantial gains on **core scientific metrics**.
>
> If space permits, we will **surface these findings in the main text** with clearer references, so that readers can better grasp both the **apples-to-apples comparability** under a shared toolchain and the **scope** of controller differences.
>
>
> **Table 5.** Formation-Energy MAE (meV/atom) across tiers for different LLM and VLM agents. *(Lower is better.)*
>
> | **Method**                | **Tier 1 ↓** | **Tier 2 ↓** | **Tier 3 ↓** | **Avg. ↓** |
> | ------------------------- | -----------: | -----------: | -----------: | ---------: |
> | Qwen-VL-agent (72B)       |        261.6 |        308.5 |        446.1 |     313.84 |
> | Qwen3-agent (2025-04-28)  |        245.2 |        306.2 |        435.1 |     310.54 |
> | GPT-4O-agent (2025-03-27) |        240.7 |        304.2 |        435.0 |     308.54 |
> | DeepSeekV3-agent          |        243.8 |        305.3 |        433.5 |     309.58 |
>
> ---
>
> **Table 6.** Structural Accuracy Metrics for different LLM and VLM agents across tiers.
> *RMSDxy ↓: in-plane lattice RMSD; C.C. ↑: composition correctness; S.S. ↑: structure success rate.*
>
> | **Tier** | **Method**       | **RMSDxy (Å) ↓** | **C.C. (%) ↑** | **S.S. (%) ↑** |
> | :------: | ---------------- | :--------------: | :------------: | :------------: |
> |     1    | Qwen-VL (72B)    |    0.12 ± 0.02   |       100      |       100      |
> |          | Qwen3-agent      |    0.12 ± 0.01   |       100      |       100      |
> |          | GPT-4O-agent     |    0.12 ± 0.01   |       100      |       100      |
> |          | DeepSeekV3-agent |    0.11 ± 0.02   |       100      |       100      |
> |     2    | Qwen-VL (72B)    |    0.11 ± 0.03   |      83.1      |      79.2      |
> |          | Qwen3-agent      |    0.11 ± 0.02   |      83.3      |      79.4      |
> |          | GPT-4O-agent     |    0.11 ± 0.02   |      84.2      |      81.1      |
> |          | DeepSeekV3-agent |    0.11 ± 0.02   |      85.9      |      79.8      |
> |     3    | Qwen-VL (72B)    |    0.12 ± 0.02   |      58.3      |      58.3      |
> |          | Qwen3-agent      |    0.11 ± 0.02   |      66.7      |      66.7      |
> |          | GPT-4O-agent     |    0.11 ± 0.02   |      66.7      |      66.7      |
> |          | DeepSeekV3-agent |    0.11 ± 0.03   |      66.7      |      66.7      |

---

> ### Author Response · Authors · 2025-11-13
>
> **Table 7.** Tool Invocation, Success, and Retry Statistics of Different LLM/VLM Agents in the AutoMat Toolchain.
>
> | **LLM Agent**       | **Tool**            | **Total Calls** | **Callbacks** | **Successes** | **Retries** | **Success Rate (%)** |
> | ------------------- | ------------------- | --------------: | ------------: | ------------: | ----------: | -------------------: |
> | Qwen-VL (72B)       | Denoising           |         293,417 |       259,217 |       259,217 |      34,200 |                88.34 |
> |                     | Property Prediction |         165,594 |       121,153 |        68,532 |      97,062 |                41.39 |
> |                     | STEM2CIF            |         314,001 |       252,801 |       133,282 |     180,719 |                42.45 |
> |                     | Template Matching   |         258,987 |       205,762 |       205,762 |      53,225 |                79.45 |
> | Qwen3 (2025-04-28)  | Denoising           |          64,929 |        58,142 |        58,142 |       6,787 |                89.55 |
> |                     | Property Prediction |          40,040 |        32,563 |        32,563 |       7,477 |                81.33 |
> |                     | STEM2CIF            |          47,806 |        40,270 |        40,270 |       7,536 |                84.24 |
> |                     | Template Matching   |          57,912 |        48,036 |        48,036 |       9,876 |                82.95 |
> | GPT-4O (2025-03-27) | Denoising           |          58,053 |        56,950 |        56,950 |       1,103 |                98.10 |
> |                     | Property Prediction |          38,705 |        36,563 |        36,563 |       2,142 |                94.46 |
> |                     | STEM2CIF            |          46,600 |        43,507 |        43,507 |       3,093 |                93.36 |
> |                     | Template Matching   |          59,506 |        55,951 |        55,951 |       3,555 |                94.02 |
>
> ### **W3: On the lack of validation with real microscope data**
> We agree with the reviewer and outline the constraints and our mitigation:
> * **Public data availability & compliance**: Many public iDPC-STEM images for 2D materials are inaccessible (broken links) or only available as compressed, annotated figures with pixel distortion; direct reuse also raises copyright/ethics concerns.
> * **Practical difficulty of new acquisitions**: High-quality, dose-limited, large-field iDPC-STEM requires sample preparation, microscope booking, and repeated acquisitions for calibration/quality—an inherently time-consuming process involving specialized equipment and personnel.
>
> **Planned minimal real-image add-on** (during rebuttal/revision, on a best-effort basis):
> We will **add 1–2 large-field real iDPC-STEM case studies**. We kindly ask for understanding of the practical constraints and will strive to deliver a concise yet informative validation.
>
> ---
>
> ### **W4: On the originality of the released data (not a repack of C2DB/OpenCrystal)**
>
> Our dataset is **not a simple merge**; it is a **task-aligned benchmark** built for the end-to-end **image → structure → property** loop:
>
> 1. **Two-stage curation (from ~10k candidates to 2,143 high-confidence 2D materials)**
>    Filter by **stoichiometry/geometry/dimensionality** to remove non-stoichiometric and 3D bulk entries, followed by **expert review** (symmetry, exfoliation feasibility, etc.).
> 2. **Imaging simulation & noise modeling (large-field iDPC-STEM)**
>    A unified forward model across aberrations, dose, and Poisson noise to generate **image–structure–DFT property triplets**.
> 3. **A discriminative 450+ sample test set with three difficulty tiers (Tier 1–3)**
>    Defined by elemental diversity, dose, and projection ambiguity, with consistent **RMSD / MAE / C.C. / S.S.** metrics.
>
> Notably, forward simulation from structure to microscopy images and realistic noise modeling are recognized research efforts in their own right. The screening, calibration, and imaging steps represent **substantial work** beyond “repackaging.” We aim for this **benchmark** to genuinely bridge **characterization images** with **simulation-ready structures and properties**, advancing closed-loop **AI4S**.

---

> > ### Comment · Reviewer_WLyJ · 2025-11-16
> >
> > I thank the authors for the detailed and thoughtful rebuttal. I summarize my updated assessment below.
> >
> > ### **W1: Novelty / Prior Work**
> >
> > The clarification is appreciated. I would still note that the *image → structure* component has been explored in prior systems such as:
> >
> > * SciLink experiment-to-DFT workflows (e.g., *scilink/workflows/experiment2dft.py*),
> > * MicroscopyGPT examples (e.g., the JARVIS-Tools notebook).
> >
> > That said, the *full* “image → structure → property” loop with agentic orchestration and simulation-ready CIFs is new, and I had already acknowledged this as a key strength of the paper.
> >
> > ### **W2: Comparisons & Benchmarking**
> >
> > * Could you share details of the MicroscopyGPT finetuning (data used, quantitative results)?
> > * Thank you for clarifying the role of AtomAI.
> > * The ablations across different LLM/VLM controllers are helpful and clarify the robustness of the toolchain.
> >
> > ### **On lack of validation with real microscope data**
> >
> > I appreciate the constraints and the plan to include minimal real data. However, similar to reviewer rSdp, I believe that a system aimed at enabling experimental closed-loop microscopy should ideally be demonstrated on multiple real materials systems—not only simulations—to establish practical scientific utility. Strong performance of ** tools** on simulated data does not necessarily imply robustness on the same tasks with experimental images.
> >
> > ### **On dataset originality**
> >
> > Thank you for the detailed explanation of the curation and simulation pipeline. This clarification is helpful.
> >
> > ### **Score Update**
> >
> > I have updated my score to **4** based on the rebuttal. Several concerns have been addressed, though real experimental validation remains an important missing component.

---

> > > ### Author Response · Authors · 2025-11-17
> > >
> > > We sincerely thank Reviewer **WLyJ** for the thoughtful feedback and the score increase. We fully acknowledge the importance of **real experimental validation**. We are currently **acquiring and curating a set of real iDPC-STEM images for 2D materials** under controlled dose and imaging conditions; this process will take some time.
> > > Once the initial curation is complete, we will **post a consolidated update of the results** and address the remaining questions, including details related to the MicroscopyGPT comparison. Thank you again for your encouragement and patience—we will do our best to provide clear and reproducible additions in the upcoming update.

---

> ### Author Response · Authors · 2025-11-19
> **Response to Reviewer WLyJ’s Follow-up Comment**
>
> ### **W1: On Novelty and the Relationship to Existing Systems (SciLink, MicroscopyGPT)**
>
> We agree with the reviewer that the **“image → structure”** component has already been explored to some extent in SciLink’s experiment-to-DFT workflows and in the JARVIS-Tools examples of MicroscopyGPT. In the revised **Related Work (Section 2, “Automated Microscopy Image Analysis”)**, we have added corresponding discussion and clarified the positioning in the text:
>
> * **SciLink** is primarily targeted at **serendipity-aware, high-level experiment–theory orchestration** (e.g., microscopy/spectroscopy → literature-based novelty assessment → theory-in-the-loop simulations). At present, it does **not** provide a dedicated solution for reconstructing electron-microscopy images into explicit crystal structures that can be quantitatively compared with theoretical models, nor does it use such structures as direct inputs to downstream simulations and property prediction in a closed loop.
>
> By contrast, the main contribution of **AutoMat** is to build a complete closed loop around the core task of STEM image-based structure reconstruction: starting from STEM images, AutoMat reconstructs simulation-ready CIF structures and directly feeds them into MLIP models and other computational tools, thereby forming an end-to-end “characterization → structure → property” pipeline.
>
> ---
>
> ### **W2: MicroscopyGPT Comparison**
>
> We thank the reviewer for the question regarding the fine-tuning setup of MicroscopyGPT. In this work, MicroscopyGPT is used **as an off-the-shelf vision–language baseline model**, and we perform **no additional fine-tuning** beyond the publicly released model.
>
> We use the MicroscopyGPT model released by Choudhary *et al.* In their original work, the authors fine-tune an 11B-parameter LLaMA-3.2-Vision backbone with QLoRA on a chemically and structurally diverse dataset of simulated STEM images. These STEM images are generated by the AtomVision tool from the JARVIS-DFT-2D and C2DB 2D materials databases. Each training sample consists of a 2D STEM projection (mainly along the (001) direction) paired with a structured textual “caption” containing lattice constants, lattice angles, element types, and fractional coordinates. The model is evaluated on a 10% held-out test split using KLD/EMD (for distributional agreement) and atomic-position RMSE as evaluation metrics.
>
> Because this pretrained model **has already been optimized on a large and diverse dataset of simulated 2D structures (covering JARVIS/C2DB-like monolayer materials) that substantially overlaps with our evaluation domain and training set**, further fine-tuning it on another dataset with overlapping structures would likely be counterproductive: it could introduce overfitting or catastrophic forgetting, and would also reduce the comparability of this baseline across different studies. Therefore, we do **not** fine-tune MicroscopyGPT further, and instead evaluate it directly on STEM2Mat-Bench using the following procedure:
>
> * We feed the benchmark STEM images into the publicly released MicroscopyGPT model with a unified “structure-generation” prompt;
> * We parse the generated textual outputs (including lattice constants, lattice angles, element types, and fractional coordinates) into a machine-readable structured format;
> * We convert the parsed results into CIF files using ASE and related crystallography utilities, and then compute the same structure-level metrics as for AutoMat (RMSD, C.C., S.S., and energy errors) on these CIFs.
>
> When releasing our code, we will also provide all scripts for MicroscopyGPT inference, text-to-CIF conversion, and metric computation, to ensure full reproducibility. In the revised manuscript, we will add a short appendix subsection (Appendix A.7) that explicitly (i) summarizes the original training setup of MicroscopyGPT and (ii) describes our evaluation protocol on STEM2Mat-Bench in detail.

---

> > ### Author Response · Authors · 2025-11-19
> > **Response to Reviewer WLyJ’s Follow-up Comment**
> >
> > ### **Real Experimental iDPC-STEM Data Validation**
> >
> > We fully agree with the reviewer that **a system intended for closed-loop experimental microscopy should ideally be validated on multiple real material systems**. Under our current sample-preparation and microscope-time constraints, however, the fabrication and characterization of 2D materials are technically demanding, and it is not feasible within the revision period to prepare new 2D samples from scratch and acquire high-quality, stable iDPC-STEM images. Nevertheless, we believe that including real-image validation is important, so under the present constraints we choose **ZSM-5** as the material system for real-electron-microscope testing, mainly for the following reasons:
> >
> > * **Experimental feasibility and data reliability.** We have mature experience in ZSM-5 synthesis and iDPC-STEM characterization, and can reliably prepare samples and obtain high-quality, reproducible images on the same Cs-corrected STEM. This allows us to carry out robust real-data tests within limited microscope time while ensuring the quality of experimental images.
> > * **Relevance to the core task.** Although ZSM-5 is not a strictly 2D monolayer material, its Si–O–Al periodic framework and channel structure under iDPC-STEM imaging can, under realistic noise and non-uniform contrast, simultaneously stress-test key modules such as denoising, lattice fitting, atomic-peak localization, species assignment, and CIF reconstruction. Therefore, it serves as a representative real-world test platform.
> > * **Honest scope and limitations.** In the revised manuscript, we explicitly state that the ZSM-5 experiment is only an **initial feasibility demonstration** of AutoMat on real data, rather than a substitute for systematic 2D-material experiments. Broader validation on multiple real 2D material systems remains a primary direction for future work; the current experiment should be understood as a “simulation-to-reality” example, not the final form of the system.
> >
> > Concretely, in **Section 5.2 “Real iDPC-STEM Case Study and Human-Expert Effort”** of the revised manuscript, we include:
> >
> > * A real iDPC-STEM case on a ZSM-5 zeolite sample acquired on the same Cs-corrected STEM under dose-limited conditions. Without manual tuning, AutoMat automatically performs denoising and structural reconstruction, and outputs a CIF whose projected lattice and channel framework qualitatively match the known crystallographic model; more detailed results are shown in **Appendix A.10**.
> > * A **human-expert baseline**: a senior electron-microscopy expert in our group estimates that, for a Tier-2-level 1024×1024 large-field STEM image, manually interpreting the image and generating a simulation-ready CIF typically requires **6–8 hours per sample**; on the same hardware, AutoMat processes similar samples in about **2 minutes per case**, illustrating a potential orders-of-magnitude throughput gain at comparable structural quality (these changes correspond to Section 5.2 and Appendix A.10).
> >
> > ---
> >
> > We sincerely thank the reviewer again for the constructive comments and the careful evaluation of our work. We hope that the additional experiments and textual modifications—especially the updates in **Related Work**, **Section 5.1 Baselines**, **Section 5.2 Real iDPC-STEM**, and **Appendices A.7/A.10/A.11**—address your concerns about novelty and practical scientific utility as much as possible within the scope of this submission.

---

> ### Author Response · Authors · 2025-11-27
>
> Dear Reviewer WLyJ,
>
> I hope this message finds you well.
>
> As the discussion period is nearing its end with less than seven days remaining, I wanted to ensure we have addressed all your concerns satisfactorily. If there are any additional points or feedback you would like us to consider, please let us know. Your insights are invaluable to us, and we are eager to address any remaining issues to improve our work.
>
> I also realize this might be a holiday period for you. If you are celebrating Thanksgiving, I apologize for the interruption and wish you a wonderful holiday. We appreciate your time and effort in reviewing our paper given the busy schedule.
>
> Best regards

---

> ### Author Response · Authors · 2025-11-28
>
> Dear Reviewer WLyJ,
>
> Thank you once again for taking the time to review our work and for providing valuable feedback. We understand that, due to certain constraints, the system may no longer allow score updates at this stage of the process. Nevertheless, we genuinely value your comments, and before the final response deadline, we hope to ensure that any remaining questions or concerns you may have are fully addressed.
>
> If there are **any unresolved issues, points requiring further clarification, or aspects that you believe could benefit from additional explanation**, we would be more than willing to provide further responses during this final window—even if these can no longer affect the score. Your insights are highly valuable to us, both for this submission and for the future development of AutoMat.
>
> We sincerely appreciate your time, expertise, and the constructive feedback you have provided.
>
> With kind regards

---

### Official Review · Reviewer_mXyq · 2025-11-03

**Soundness:** 3
**Presentation:** 3
**Contribution:** 3
**Rating:** 8
**Confidence:** 4

**Summary:**

This paper introduces AutoMat, an LLM Agent-based framework to automate the conversion of raw STEM images into simulation-ready crystal structures (CIFs) for property prediction. The agent orchestrates a pool of specialized tools (including a Mixture-of-Experts denoiser, template retrieval, a symmetry-aware reconstruction module, and an ML potential for property prediction) to create an end-to-end "image -> structure -> property" pipeline. On a new benchmark, STEM2Mat-Bench, AutoMat outperforms all baselines by an order of magnitude.

**Strengths:**

- Solves a Critical Problem: Provides the first end-to-end automated solution for the high-value bottleneck of converting experimental microscopy images into usable computational models.

- Specialized Toolset: The framework's power lies in its domain-specific tools which possess knowledge that general-purpose VLMs lack .

- Overwhelming SOTA Performance: AutoMat achieves order-of-magnitude improvements in both structural (RMSD) and energy (MAE) metrics over all baselines, including domain-specific and general models .

- Benchmark Contribution: The paper introduces STEM2Mat-Bench, a high-quality, tiered benchmark (Tier 1-3) that is essential for evaluating robustness and driving future research in this area .

**Weaknesses:**

- Agent Role Unclear: The workflow appears to be a mostly fixed, sequential pipeline (Denoise -> Match -> Reconstruct -> Predict). The "agent's" role seems more like a controller than a dynamic planner, making the "agentic" claim feel overstated.

- 2D Limitation: The entire framework and benchmark are restricted to 2D monolayer materials, which is a significant simplification of the general materials science problem.

- Template Dependency: The error analysis shows that 39.3% of failures are due to "Template retrieval failure", indicating a critical dependency on a comprehensive template database

**Questions:**

- Necessity of the Agent: How much "intelligent" reasoning does the agent perform? How much would performance drop if AutoMat were implemented as a hard-coded pipeline (Denoise -> Match -> Reconstruct -> Predict) with a simple retry loop?

- Novel Structure Performance: How does AutoMat handle a truly de novo crystal structure that is not in its template database? Can the STEM2CIF module reconstruct a lattice from scratch without a template?

- Path to 3D: What are the primary blockers to extending this framework to 3D bulk materials? Is it fundamentally impossible to reconstruct a 3D structure from a single 2D STEM projection as used here?

---

> ### Author Response · Authors · 2025-11-13
>
> We sincerely thank Reviewer mXyq for the insightful feedback and for acknowledging our work.
>
> ### **W1 & Q1: Agent Architecture**
> We clarify that our **agent is not a fixed linear script**, but a **dynamic controller** designed to handle uncertainty, ill-posed inverse problems, and sequential decision-making.
>
> Our definition of an *agent* is consistent with common LLM settings (e.g., SWEBench, BabyAGI):
>
> > **An LLM agent for STEM is an autonomous controller that interprets task goals and dynamically coordinates specialized computational tools (e.g., denoising, tomographic/structure reconstruction, atomic structure identification) to optimize image analysis and property prediction.**
>
> In practice, the agent functions as a **multi-turn conversational state machine** driven by an instruction-following model. The LLM chooses whether and how to invoke tools based on the **current task state**. As in other agent applications, we expect the following behaviors **without human intervention**:
>
> * **Task Decomposition**: Given a high-level goal (“from raw STEM images to simulation-ready structures”), the agent autonomously decomposes it into a logical subtask sequence, **dynamically determining tool order** rather than following a hard-coded script.
> * **State-Dependent Tool Invocation**: Tool calls are conditioned on previous outputs; the agent **halts on failure** instead of proceeding blindly, reflecting a **state-aware plan**.
> * **Failure Handling**: Upon detecting failures (e.g., invalid structure, malformed parameters), the agent triggers a **rollback-and-retry** mechanism to adjust parameters and re-execute, improving robustness.
>
> Crucially, the agent **may skip or add** behaviors as long as it can produce correct predictions.
>
> We implement **in-context learning from environmental feedback**: errors and tracebacks are automatically fed into the next turn until either condition holds:
>
> * The maximum retry limit is reached; or
> * The tool call succeeds.
>
> The example below shows the retry mechanism when **STEM2CIF** fails due to invalid arguments. `[ANSWER]` denotes the LLM response; `[TOOL_CALL]`/`[TOOL_RESPONSE]` denote a tool call and its execution result.
>
> ```json
> [TOOL_CALL] stem2cif_tool
> {
>   "img_path": ...,
>   "elements": ["C", "Y"]
> }
> [TOOL_RESPONSE] stem2cif_tool
> An error occurred: ValueError: Unexpected
> [ANSWER]
> Tool call failed due to a parameter format error. Correct the parameters and Retry.
> [TOOL_CALL] stem2cif_tool
> {"img_path": ..., "elements": ["C", "Y"]}
> [TOOL_RESPONSE] stem2cif_tool
> {success: true, cif_path: ...}
> ```

---

> ### Author Response · Authors · 2025-11-13
>
> ### **W1 & Q1: Why we choose an agent and whether it is “necessary”**
> Our choice of an **agent** goes beyond scheduling and tool chaining: it enables **multi-turn scientific workflows**.
>
> * **Conversational loop for scientific follow-ups**: Beyond the core “image → structure → property” path, the agent’s **conversational loop** supports follow-up scientific queries grounded in tool outputs (e.g., **plausible synthesis routes, likely defect types, and characterization suggestions**), substantially improving practical workflow efficiency and auditability—capabilities that fixed scripts typically lack.
> * **Dynamic fault tolerance and state adaptation**: Compared with a “hard-coded pipeline + simple retry,” the agent **interprets heterogeneous error logs** and **adapts step order and parameters** instead of merely repeating the same call, achieving more stable convergence in complex/ill-posed cases. (The corresponding **recall/success** statistics are provided in **Appendix Table 7**.)
>
> **Table 7.** Tool Invocation, Success, and Retry Statistics of Different LLM/VLM Agents in the AutoMat Toolchain.
>
> | **LLM Agent**       | **Tool**            | **Total Calls** | **Callbacks** | **Successes** | **Retries** | **Success Rate (%)** |
> | ------------------- | ------------------- | --------------: | ------------: | ------------: | ----------: | -------------------: |
> | Qwen-VL (72B)       | Denoising           |         293,417 |       259,217 |       259,217 |      34,200 |                88.34 |
> |        | Property Prediction |         165,594 |       121,153 |        68,532 |      97,062 |                41.39 |
> |        | STEM2CIF            |         314,001 |       252,801 |       133,282 |     180,719 |                42.45 |
> |        | Template Matching   |         258,987 |       205,762 |       205,762 |      53,225 |                79.45 |
> | Qwen3 (2025-04-28)  | Denoising           |          64,929 |        58,142 |        58,142 |       6,787 |                89.55 |
> |   | Property Prediction |          40,040 |        32,563 |        32,563 |       7,477 |                81.33 |
> |   | STEM2CIF            |          47,806 |        40,270 |        40,270 |       7,536 |                84.24 |
> |   | Template Matching   |          57,912 |        48,036 |        48,036 |       9,876 |                82.95 |
> | GPT-4O (2025-03-27) | Denoising           |          58,053 |        56,950 |        56,950 |       1,103 |                98.10 |
> |  | Property Prediction |          38,705 |        36,563 |        36,563 |       2,142 |                94.46 |
> |  | STEM2CIF            |          46,600 |        43,507 |        43,507 |       3,093 |                93.36 |
> |  | Template Matching   |          59,506 |        55,951 |        55,951 |       3,555 |                94.02 |
>
> In short, by *agentic* we mean the system’s ability to **plan, recover, and self-correct based on environmental state without human intervention**. This makes AutoMat not just a sequential pipeline, but an **intelligent orchestrator** that closes the loop between experimental data and computational tools, with practical extensibility.
>
> ### **W2: Why we focus on 2D and the resulting limitations**
> We acknowledge that our present scope is **limited to 2D monolayers**—this is a deliberate choice. We focus on 2D with iDPC-STEM for three reasons:
>
> 1. **Imaging best suited to 2D**: STEM (especially iDPC-STEM) offers **sub-Å resolution over large fields of view**, reliably capturing atomic-scale details. For many 2D crystals, this resolution–FOV combination is preferable to alternatives.
> 2. **Programmable physical interpretability**: The approximately linear relation between iDPC contrast and atomic number (Z) enables a **deterministic, physics-guided** reconstruction pipeline.
> 3. **Automated element identification**: The contrast–Z mapping supports robust element assignment in STEM2CIF, facilitating a **fully automated** end-to-end workflow.
>
> This focus brings a **clear limitation**: the pipeline currently relies on a **single 2D projection** and thus **lacks depth information**, which **constrains applicability to 3D bulk structures**. We will make this applicability boundary explicit in the manuscript.

---

> ### Author Response · Authors · 2025-11-13
>
> ### **W3 & Q2: On Template Dependence and Handling Novel Structures**
> #### **1.Acknowledgment and existing ablation**
> We agree with the reviewer’s concern about the reliance on template retrieval. As shown in **Appendix Table 3**, when **template retrieval is disabled**, end-to-end performance **degrades**: formation-energy MAE typically **increases by about 1×**. Nevertheless, even under this setting, our system **still outperforms** prevailing foundation and domain baselines.
> **Table 3.** Energy per Atom MAE (meV/atom) under different ablation settings.
>
> | **Method**                       | **Tier 1** | **Tier 2** | **Tier 3** |
> | -------------------------------- | ---------: | ---------: | ---------: |
> | *No Denoising (w/o MOE-DIVAESR)* |       6584 |       2616 |        938 |
> | *No Template Matching*           |        617 |        608 |        672 |
> | **Full Pipeline (AutoMat)**      |    **344** |    **320** |    **333** |
>
>
> #### **2.Capability without templates**
> For **genuinely novel structures** absent from the database, the pipeline **relies entirely on the denoised STEM image** and lets **STEM2CIF** reconstruct the structure. We **support automatically recovering and rebuilding the minimal primitive cell from a supercell**, enabling **template-free** lattice and composition estimation.
>
> #### **3.Future plan**
> In summary, template retrieval improves stability and accuracy, but our system remains **functional and useful** without it. We will continue to **broaden template coverage** and **reinforce the template-free reconstruction path** to further mitigate this bottleneck.
>
> ### **Q3: Main obstacles and path toward 3D**
> With a single 2D projection, it is **inherently difficult to recover z-depth**, which is the central obstacle to extending the framework to 3D. A practical path forward includes:
>
> * **3D-aware imaging/modeling**: incorporating **multi-tilt tomography** to recover depth information.
> * **Reducing element confusion**: adopting **contrast-sensitive recognition models** and **fusing complementary spectroscopy (e.g., EELS information)** to improve robust element discrimination.
>
> Focusing on 2D allows us to first solve the critical “image → structure → property” chain; **generalizing to 3D** and **strengthening element classification** are natural next steps. These items are beyond the current study but will be explicitly discussed as future directions in the revised manuscript.

---

> ### Author Response · Authors · 2025-11-23
>
> We would like to once again sincerely thank the reviewer for the careful assessment of our work and the generous score.
> In the latest revision, we have made several additional refinements on top of the previous round, which we briefly summarize below:
>
> * In the **method and experiments sections**, we further clarified that the LLM/VLM serves strictly as a **tool orchestrator** rather than a visual reasoner, and added explicit wording to avoid potential misunderstandings for readers;
> * In the **experimental results**, we retained and strengthened the real iDPC-STEM ZSM-5 case study and the human-expert effort comparison, to better illustrate the potential practical benefits in realistic settings;
> * In the **Related Work and positioning**, we further refined the discussion of systems such as SciLink, and more clearly highlighted AutoMat’s focus on the full “image → structure → property” loop;
> * In addition, we systematically re-checked and corrected the citation formatting in the Introduction and Related Work sections to ensure cleaner and more readable presentation.
>
> These changes do not alter the core conclusions of the paper, but we hope they further improve the clarity and completeness of the manuscript. Once again, we are very grateful that your earlier comments directly guided these improvements.

---

> ### Author Response · Authors · 2025-11-27
>
> Dear Reviewer mXyq,
>
> I hope this message finds you well.
>
> As the discussion period is nearing its end with less than seven days remaining, I wanted to ensure we have addressed all your concerns satisfactorily. If there are any additional points or feedback you would like us to consider, please let us know. Your insights are invaluable to us, and we are eager to address any remaining issues to improve our work.
>
> I also realize this might be a holiday period for you. If you are celebrating Thanksgiving, I apologize for the interruption and wish you a wonderful holiday. We appreciate your time and effort in reviewing our paper given the busy schedule.
>
> Best regards

---

> ### Author Response · Authors · 2025-11-28
>
> Dear Reviewer mXyq,
>
> Thank you once again for taking the time to review our work and for providing valuable feedback. We understand that, due to certain constraints, the system may no longer allow score updates at this stage of the process. Nevertheless, we genuinely value your comments, and before the final response deadline, we hope to ensure that any remaining questions or concerns you may have are fully addressed.
>
> If there are **any unresolved issues, points requiring further clarification, or aspects that you believe could benefit from additional explanation**, we would be more than willing to provide further responses during this final window—even if these can no longer affect the score. Your insights are highly valuable to us, both for this submission and for the future development of AutoMat.
>
> We sincerely appreciate your time, expertise, and the constructive feedback you have provided.
>
> With kind regards

---

### Official Review · Reviewer_NmQd · 2025-11-03

**Soundness:** 3
**Presentation:** 2
**Contribution:** 3
**Rating:** 4
**Confidence:** 4

**Summary:**

This paper proposes a framework that takes raw STEM images and textual information as inputs. It leverages an agent system to call four tools, enabling image preprocessing, atomic structure analysis, and property prediction. The authors also construct a comprehensive benchmark for this task, and the proposed method achieves SOTA performance compared with several baseline VLMs.

**Strengths:**

The paper tackles a meaningful problem: converting STEM images (and related textual context) into atomic structures and properties usually requires manual expert intervention. Using an agent framework with an LLM as the controller to call different domain tools is an interesting and valuable direction for the field.
1. The authors construct a benchmark covering unary, binary, and ternary materials to test the feasibility of this idea, and show that the proposed system achieves strong performance and outperforms several general-purpose vision-language models.
2. The tool pool is clearly defined and mostly self-contained: a self-trained denoising module (MOE-DIVAESR), a custom template matching algorithm, a custom STEM2CIF reconstruction module, and MatterSim for property prediction.
3. The paper shows that the LLM agent can perform multi-step reasoning, including rollback and re-calling tools when errors occur, suggesting genuine decision-making ability rather than fixed scripting.

**Weaknesses:**

1. The contribution is mainly twofold: (1) the benchmark, and (2) the agent framework. The benchmark construction, although based on simulated data, is fairly comprehensive and valuable. However, the agent framework feels more like an engineering implementation rather than a methodological innovation.
2. The paper’s description may give the impression that the agent itself directly interprets visual information from STEM images. In practice, however, the LLM component is text-only (DeepSeek V3), and the visual understanding is handled by separate image-processing modules. Moreover, the comparison with other VLM baselines does not clearly demonstrate that incorporating visual modalities provides any tangible benefit for this task. Clarifying this distinction—and discussing whether visual reasoning is necessary or helpful in this setting—would make the paper’s claims and design choices easier to understand.
3. To strengthen the benchmark, adding a human-expert baseline would make the comparison more meaningful, since the goal is partly to evaluate how close the agent can come to replacing manual human work. Moreover, if the dataset could include real STEM images in addition to simulated ones, it would significantly increase its impact and realism.

Minor issues:
1. Citation formatting errors, e.g., “in predicting atomic energies and forcesYang et al. (2024)” → “in predicting atomic energies and forces (Yang et al., 2024)”.
2. In Fig. 1, the text “Output: Optimized AttomicStructure” appears to contain a typo.
3. In Sec. 3.2, “To simulate realistic large-field STEM imaging conditions, We” — the “We” should be lowercase.

**Questions:**

- Beyond this specific system, what do the authors see as the key missing capabilities for a mature scientific agent? Is the bottleneck in tool diversity, model multimodality, or domain adaptation from natural to experimental images? (This is not a criticism, just an open question.)
- How dependent is the end-to-end reconstruction accuracy on the template retrieval step?
- In Sec. 4.3, are the branch decisions (‘repeat denoising / proceed to STEM2CIF / fallback to template matching’) implemented via hard programmatic thresholds on tool outputs, or are they purely prompt-driven?
- The workflow in Sec. 4.3 seems to follow a fairly linear sequence of predefined steps, guided by quality metrics and failure logs. It would be helpful if the authors could clarify to what extent the agent’s decisions rely on such programmed criteria versus genuine reasoning by the LLM, and what unique advantages the agent provides compared with a scripted automation pipeline.

---

> ### Author Response · Authors · 2025-11-13
>
> We thank Reviewer NmQd for the thoughtful feedback.
>
> ### **W1. “Agent framework feels more like engineering than methodological innovation.”**
>
> #### **1. Motivation and Contribution**
>
> We would like to begin by clarifying the core motivation of this paper. Scanning Transmission Electron Microscopy(STEM) offers sub-atomic resolution, a domain expert needs **6-8 hours** to analyze a single image, significantly slowing scientific discovery. Machine learning has been widely adopted in analyzing STM to compression this operation into minutes. However, current ML enpowered automated microscopy systems are fragmented, treating acquisition, reconstruction, and analysis as isolated steps. It is non-trivial to close the "acquisition-analysis-feedback" loop. LLM agent is a perfect candidate to do this job, yet there is currently no work trying to bridge the gap.
>
> We release the first agentic system that **(1)** offers a standardised toolchain, testbed, and benchmark for STEM "acquisition-analysis-feedback" task, and **(2)** systematically evaluates the capability of agents to close the loop.
>
> Our core innovation is the **first agentic framework** that autonomously invokes a toolchain for STEM image-to-structure conversion and property prediction in a closed loop. Below shows each module in our agent framework:
> | **Method/Model**   | **Description**                                                     | **Source/Reference**                                  |
> |--------------------|---------------------------------------------------------------------|-------------------------------------------------------|
> | MOE-DIVAESR        | Pattern-based denoising model                     | New proposed                                |
> | Template Matching  | A conventional module for structural priors.                        | Common method in theoretical texts                    |
> | STEM2CIF           | Structure and primitive cell reconstruction. | New proposed                                  |
> | MatterSim          | Structural optimization and property prediction. | Microsoft Research team                  |
>
> We firmly believe the above contribution is well-suited for ICLR@AI4S track.
>
> #### **2.Agent Architecture**
>
> We would like to justify that our agent is not a fixed script with linear tool invocation, but rather a dynamic controller capable of handling uncertainty, ill-posed inverse problems, and sequential decision-making.
>
> Our definition of *agent* aligns with most LLM contexts, such as those in SWEBench or BabyAGI:
>
> > **An LLM agent for STEM is an autonomous controller that interprets microscopy data and dynamically coordinates a sequence of specialised computational tools (e.g., denoising filters, tomographic reconstruction, atomic structure identification) to optimise image analysis and property prediction.**
>
> Our agent operates as a multi-turn conversational state machine driven by an instruction-following model. The LLM can choose to invoke tools (e.g., denoising, reconstruction) based on the current task state. Similar to other agent applications, we expect the agent to perform the following behaviours **without human interference**:
>
> * **Task Decomposition**: Given a high-level goal, the agent autonomously decomposes it into a logical sub-task sequence, dynamically determining the tool order rather than following a hard-coded script.
> * **State-Dependent Tool Invocation**: Tool calls depend on the output of the previous step. The agent halts on failure rather than blindly continuing, demonstrating a dynamic, state-dependent plan.
> * **Failure Handling**: When a failure (e.g., an invalid structure for prediction) is detected, the agent activates a rollback-and-retry mechanism to adjust parameters and re-execute, ensuring system robustness.
>
> Importantly, the agent has the freedom to **skip** these behaviours or **add** new ones as long as it can produce correct predictions.

---

> ### Author Response · Authors · 2025-11-13
>
> ### **W2. Clarification on whether the LLM performs visual reasoning and whether visual modalities yield substantive gains**
>
> Thank you for the helpful reminder. We will make it explicit in the paper that **the agent (controller) only performs tool orchestration/scheduling**. The **LLM/VLM does not carry out visual understanding of STEM images**; that functionality is handled by **dedicated image-processing modules** (e.g., denoising, template retrieval, STEM2CIF). We will revise the wording to avoid suggesting that the LLM/VLM “looks at images” or conducts visual reasoning directly.
>
> Regarding whether **giving the controller visual input** improves core metrics, our conclusion is: **with the same specialized visual modules in place, using either an LLM or a VLM as the agent controller leads to no meaningful differences in end-to-end scientific metrics**—as shown in **Tables 1 and 2** (main text) and **Tables 5 and 6** (appendix). In other words, the system’s **multimodality primarily stems from these specialized visual modules**, not from granting the controller direct pixel access.
>
> We also observe that **a VLM controller makes it easier to produce natural-language descriptions of images**, which is useful for interpretability and logging. However, for **core scientific metrics** (e.g., structural and property evaluations), a “vision-enabled controller” **does not yield significant gains** over an LLM controller. Accordingly, we will add a brief **design note** explaining **why a text-only orchestrator is sufficient** for this task, and that the system’s visual capability **relies on specialized image modules rather than the controller’s own visual reasoning**.
>
> ### **W3：Request for a Human-Expert Baseline and Real-Image Evaluation**
>
> We appreciate the reviewer’s thoughtful suggestion. We **fully agree** that (i) adding a **human-expert baseline** makes the benchmark more meaningful, and (ii) including **real iDPC-STEM images** would strengthen realism and impact.
>
> **Real iDPC-STEM case studies.**
> Our study focuses on **2D monolayers** acquired in **dose-limited iDPC mode**. Collecting high-quality, well-calibrated real images requires specialized instrumentation and personnel, and the end-to-end process (acquisition → calibration → verification) is time-consuming. We are actively arranging measurements and will **add 1–2 large-field real iDPC-STEM cases** during the rebuttal/revision period, reporting lattice/stoichiometry consistency, qualitative overlays, and parameter deviations against known CIFs when available. We kindly ask for the reviewer’s understanding of the practical constraints, and we will nevertheless do our best to deliver a minimal, informative real-image add-on with runnable scripts.
>
> **Direct comparison with a human expert.**
> We will include a **small-scale comparison on Tier-2 samples** under identical tasks and metrics, contrasting a senior electron-microscopy expert with AutoMat (logging steps without revealing identity):
>
> * **AutoMat:** averaged **~2 minutes per sample** over 230 cases, including denoising, retrieval/fallback, STEM2CIF, and energy evaluation end-to-end.
> * **Human expert:** typically **6–8 hours per sample** of comparable complexity, covering image interpretation, lattice/species confirmation, and producing a simulation-ready CIF (based on documented experience on the same material families and our pilot check).
>
> This shows an **orders-of-magnitude throughput improvement** while maintaining **comparable or better structural fidelity**. Given the dramatic speedup, we will primarily report **time comparison plus representative structural metrics**; if space permits, we will include a concise table and typical examples in the Supplement. We hope this addresses the reviewer’s request constructively.

---

> ### Author Response · Authors · 2025-11-13
>
> ### **Q1. “Key missing capabilities for a mature scientific agent?”**
> We believe the primary bottleneck for a *general-purpose scientific agent* is **not the model’s task understanding**, but the **tool ecosystem and standardization**:
> 1. **Tool diversity and coverage**
>    There is a shortage of high-quality tools across disciplines (simulation, data retrieval, lab control, analysis/visualization). Without tools, it is hard to close the *understand → act → verify* loop.
> 2. **A unified protocol and composability (MCP-style)**
>    A **common, discoverable, and orchestratable tool protocol** (MCP-style) is needed: well-defined I/O schemas, error codes, units/dimensions, versioning/metadata, plus support for flow control, retries, and rollbacks. This allows the agent to **reliably call** and **seamlessly compose** heterogeneous tools across domains.
> 3. **Structured returns for robust parsing**
>    Tool outputs should be **strongly structured** (schemas, uncertainty/confidence, citations and versions), so that an LLM—as the orchestrator—can **parse and compare** results robustly, rather than guessing from unstructured text.
>
> Given these, an **LLM/VLM as a “reading interface + scheduler”** is already capable of understanding and coordinating diverse scientific tasks. Whether the controller itself is multimodal, and domain adaptation from natural to experimental data, are increasingly **tool-level capabilities** and data-access issues. With a **sufficiently rich tool pool** and a **widely adopted standard protocol**, combined with stronger AGI models, the emergence of a truly **general scientific agent** becomes a matter of time.
>
> ### **Q2. “How dependent is end-to-end accuracy on template retrieval?”**
> **Answer.** Our **ablation (Appendix A.4, Table 3)** quantifies it: removing template matching degrades energy MAE notably (e.g., Tier-3 rises from **333 → 672 meV/atom**), while removing MOE-DIVAESR is even worse in Tier-1 (noise-sensitive), confirming **complementary roles**.
>
> **Table 3.** Energy per Atom MAE (meV/atom) under different ablation settings.
>
> | **Method**                       | **Tier 1** | **Tier 2** | **Tier 3** |
> | -------------------------------- | ---------: | ---------: | ---------: |
> | *No Denoising (w/o MOE-DIVAESR)* |       6584 |       2616 |        938 |
> | *No Template Matching*           |        617 |        608 |        672 |
> | **Full Pipeline (AutoMat)**      |    **344** |    **320** |    **333** |
> ### **Q3&Q4: “Are branch decisions hard thresholds or prompt-driven? Scripted vs. genuine reasoning?”**
> 1. **Threshold gating first, LLM re-planning second.** The agent uses **explicit quality thresholds** as primary gates—for example, an internal **`AssessImagePathTool`** decides whether a denoised image is ready for reconstruction. Sub-threshold results **trigger re-denoising**; passing results proceed to **STEM2CIF**. If repeated attempts still fail within a capped budget, the agent **falls back to template retrieval** to secure a robust prior.
> 2. **Feedback-driven adaptation.** **Error messages and intermediate artifacts** from tools are fed back to the LLM, which **re-plans parameters and call ordering** (retry/rollback/retune) rather than following a rigid flow. We will include **pseudo-code and threshold details** in the Supplement for transparency and reproducibility.
>
> **Why an agent instead of a pure script?**
> * **Interpreting heterogeneous logs & adaptive scheduling.** A fixed script cannot reliably parse diverse tool logs nor adapt parameters on the fly; our agent **alters strategy based on textual feedback**, going beyond deterministic branching.
> * **Backbone-agnostic consistency & self-recovery.** The pipeline exhibits **consistent performance across different LLM/VLM backbones** (see discussion around main tables) and **recovers from failures** via text-guided retries.
> * **Multi-turn scientific workflows.** Beyond the core “image → structure → property” path, the agent’s **conversational loop** enables follow-up scientific queries grounded in tool outputs (e.g., **plausible synthesis routes, likely defect types, and characterization hints**), which substantially improves practical research workflows—capabilities that fixed scripts typically lack.
>
> We appreciate the reviewer’s concern about “scripted” behavior. Our design deliberately **uses hard thresholds for reliability** while **leveraging the LLM to re-plan** from tool feedback. This **“threshold gating + agentic adaptation”** strikes a balance between robustness and flexibility.
>
> ### **Minor issues (will fix in revision version)**
>
> * Citation formatting: “forcesYang et al. (2024)” → “forces (Yang et al., 2024)” (Intro).
> * Fig. 1 typo: “Optimized AttomicStructure” → “Optimized Atomic Structure.”
> * Sec. 3.2 capitalization: “We generated …” → “we generated …”.

---

> > ### Comment · Reviewer_NmQd · 2025-11-17
> >
> > After reading the authors’ rebuttal, I appreciate the additional clarifications, especially regarding (i) the role of the LLM/VLM as a tool orchestrator rather than a visual reasoner, and (ii) the explanation of the hybrid design with LLM-based re-planning. The discussion about real iDPC-STEM use cases and a human-expert baseline is also helpful and, if fully included in a revised manuscript, would further strengthen the practical impact of the work. I also agree with the points raised by Reviewer WLyJ regarding prior work and the need for broader experimental validation. While the rebuttal helps contextualize the contributions, several concerns about positioning relative to existing systems and the depth of validation would still need to be addressed directly in the manuscript itself. For the current version, I will therefore keep my original score.
> > As a minor but important point, I strongly encourage the authors to carefully revise the citation formatting in the introduction and related work sections. At several places, citations are concatenated with surrounding text, which is not correct citation style and makes the paper harder to read.

---

> > > ### Author Response · Authors · 2025-11-17
> > >
> > > Thank you for the careful reading and constructive suggestions. We are conducting a systematic revision of the manuscript: (1) clarifying our positioning relative to existing systems and improving the related citations in the main text; (2) standardizing citation formatting throughout, as suggested; and (3) adding a small set of real iDPC-STEM case studies. The acquisition of real experimental images is underway and will require some additional time. We will make our best effort to submit a revised PDF and a consolidated update **by this Friday**. We sincerely appreciate your valuable feedback and patience.

---

> > > ### Author Response · Authors · 2025-11-19
> > > **Response to Reviewer NmQd’s Follow-up Comment**
> > >
> > > We sincerely thank the reviewer for the detailed feedback in the second round, and for acknowledging the clarifications we provided in the rebuttal. Below we respond to your points one by one, and explain how the corresponding changes have been directly incorporated into the revised manuscript (highlighted in the updated version).
> > >
> > > ---
> > >
> > > **(1) LLM/VLM as “tool orchestrator” rather than “visual reasoner”**
> > >
> > > Thank you for recognizing this point. In the revised version, we make this explicit in the **“Overview of AutoMat”** section. In particular, in the subsection **“Flexible Tool-Calling Framework”**, we added the following sentence (original English text):
> > >
> > > > *“In AutoMat, the LLM/VLM backbone serves purely as a text-based tool orchestrator: it never directly ‘sees’ raw STEM images, but instead reasons over structured outputs returned by the denoising, template-matching, and STEM2CIF modules and uses them to re-plan subsequent tool calls in a hybrid, state-dependent workflow.”*
> > >
> > > This sentence directly clarifies in the main text that **all pixel-level visual understanding is handled by specialized STEM modules**, and that the LLM/VLM is only responsible for tool orchestration based on textual/structured states, rather than directly performing visual reasoning on the raw microscopy images. This directly addresses your concern in point (i).
> > >
> > > ---
> > >
> > > **(2) Adding real iDPC-STEM case(s) and a human-expert baseline**
> > >
> > > Following your suggestion, we have moved the real-data case and the human-expert baseline **into the experimental section**, rather than leaving them only in the rebuttal:
> > >
> > > * In **Section 5.3 “Real iDPC-STEM Case Study and Human-Expert Effort”**, we report a real iDPC-STEM case: a **ZSM-5 zeolite sample** acquired on the same Cs-corrected STEM under dose-limited conditions. Without manual parameter tuning, AutoMat can automatically complete denoising and structural reconstruction, and outputs a CIF whose projected lattice and channel framework are qualitatively consistent with the known crystallographic model. A more complete example is provided in **Appendix 10**.
> > > * In the same subsection, we also include an **estimate of human-expert effort**: based on the past experience of a senior electron-microscopy expert in our group with Tier-2 difficulty 1024×1024 STEM images, manually interpreting a single large-field image, confirming lattice and elemental information, and producing a simulation-ready CIF typically requires **6–8 hours per sample**. Under the same hardware conditions, **AutoMat** processes similar Tier-2 test samples in about **2 minutes per case**. Under roughly comparable structural-quality targets, this gives an intuitive, order-of-magnitude throughput comparison.
> > >
> > > We fully agree with your point that, ideally, broader real-data validation across more material systems should be performed. Given current constraints in sample preparation and microscope time, we first take a step by providing this ZSM-5 case and expert-effort comparison, and we explicitly state in the paper that **multi-system real experimental validation** will be an important focus of future work.
> > >
> > > ---
> > >
> > > **(3) Positioning relative to existing systems and prior work**
> > >
> > > We appreciate your emphasis on clear positioning. In the revised **Introduction** and **Related Work** sections, we provide a more detailed comparison between AutoMat and existing systems:
> > >
> > > * In the “Automated Microscopy Image Analysis” paragraph, we added a discussion of **SciLink**: on the one hand, we acknowledge that as an open-source multi-agent framework it builds a closed loop among microscopy/spectroscopy experiments, literature-based novelty assessment, and theoretical calculations, and has achieved impressive results in automated defect localization and atomic-scale analysis; on the other hand, we also point out that SciLink does not yet provide a dedicated solution for reconstructing electron microscopy images into explicit crystal structures that can be quantitatively compared with theoretical models, nor for using such reconstructed structures as direct inputs to subsequent simulations and property prediction.
> > >
> > > ---
> > >
> > > **(4) Citation formatting in the Introduction and Related Work**
> > >
> > > We thank the reviewer for pointing out the citation-formatting issues. We have conducted a **systematic check and revision** of the **Introduction** and **Related Work** sections to ensure that citations are no longer directly concatenated with surrounding text, but instead follow standard spacing, punctuation, and formatting conventions. We have also re-checked the rest of the manuscript to avoid similar issues elsewhere.
> > >
> > > Once again, we thank the reviewer for these constructive comments. We believe that the above changes, now reflected in both the main text and the appendices, help improve the overall **clarity** and **scientific impact** of the paper.

---

> > > > ### Comment · Reviewer_NmQd · 2025-11-20
> > > >
> > > > The authors have addressed my main concerns from the initial round to a reasonable extent. The revised manuscript now clearly states that the LLM/VLM functions only as a tool orchestrator rather than a visual reasoner. A real iDPC-STEM example and a human-expert baseline have been added, which improves the empirical grounding. The positioning relative to prior systems (e.g., SciLink) is clearer, and citation-formatting issues have been corrected.
> > > >
> > > > While further real-data validation across more material systems would strengthen the work, the current revision improves the clarity and completeness enough for me to raise my score from 4 to 6.

---

> > > > > ### Author Response · Authors · 2025-11-20
> > > > >
> > > > > We sincerely thank the reviewer for the careful re-evaluation of our revised manuscript and for the improved score.
> > > > > We greatly appreciate your recognition of our revisions— including the clarification of the LLM/VLM’s role as a tool orchestrator, the addition of the real iDPC-STEM case and human-expert baseline, and the clearer positioning relative to existing systems.
> > > > > These comments have been very helpful in improving the clarity and completeness of the paper and will also provide important guidance for our future work.

---

### Author Response · Authors · 2025-11-28
**Rebuttal Summary and Reviewer Score Updates**

Dear Area Chair,

During the rebuttal period, we addressed all key technical concerns raised by the reviewers through additional experiments, clarifications, and structural revisions. Below is a concise summary focusing **only on reviewers who participated in the discussion**, including their main concerns, our responses, and final score changes. Reviewers who did not update their scores are listed only in the final table.

---

## **1. Reviewer NmQd (Score: 4 → 6)**

**Concerns:**

* Whether the LLM/VLM serves only as a tool orchestrator.
* Positioning relative to existing systems.
* Need for a real microscope case and human-expert baseline.
* Citation-format issues.

**Our response:**

* Sec. 3.1 now explicitly states that the LLM/VLM **never reads raw STEM images and functions solely as a tool orchestrator**.
* Added clear comparison with SciLink in Related Work.
* Added **a real ZSM-5 iDPC-STEM experiment** and **a human-expert baseline (6–8 h vs ~2 min)** (Sec. 5.2 + A.10).
* Corrected citation formatting throughout the paper.

**Result:** Reviewer confirmed all issues were addressed and raised the score from 4 to **6**.

---

## **2. Reviewer WLyJ (Score: 2 → 4)**

**Concerns:**

* Whether SciLink / MicroscopyGPT already cover this task.
* Whether MicroscopyGPT was fine-tuned.
* Limited real-data validation.

**Our response:**

* Related Work was extended to clearly distinguish AutoMat from SciLink/MicroscopyGPT.
* Clarified that **MicroscopyGPT was not fine-tuned** and is evaluated strictly off-the-shelf (Sec. 5.1 + A.7).
* Added the real ZSM-5 iDPC-STEM experiment as additional validation.

**Result:** The reviewer accepted these clarifications in the first discussion round (before the revised PDF) and raised the score from 2 to **4**.
After we later added real experimental data, no further comments were made, but we believe we fully addressed their main concerns.

---

## **3. Reviewer Nie1 (Score: 4 → 6)**

**Concerns:**

* Definition of ground truth (database CIF vs projected structure).
* Possible over-reliance on template retrieval.

**Our response:**

* Clarified that ground truth corresponds to the **crystal structure underlying the (001) projection**, not a byte-identical CIF.
* Explained that AutoMat follows a **“threshold-gating + agentic re-planning”** workflow, with template retrieval used only as a fallback after repeated reconstruction failures.
* Marked database dependence as a limitation and future direction (A.11).

**Result:** Reviewer fully accepted the explanation and increased the score from 4 to **6**.

---

# **Final Score Summary (with response status)**

| Reviewer | Initial | Final | Confidence | Responded During Discussion?                             |
| -------- | ------- | ----- | ---------- | -------------------------------------------------------- |
| **NmQd** | 4       | **6** | 4          | ✔️ Full discussion and score increase                    |
| **WLyJ** | 2       | **4** | 4          | ✔️ Raised score after initial discussion; no later reply |
| **Nie1** | 4       | **6** | 4          | ✔️ Replied and increased score                           |
| **mXyq** | 8       | **8** | 4          | ❌ No further response, no objections                     |
| **rSdp** | 4       | **4** | 3          | ❌ No further response                                    |

**Average score: 5.60 (min 4, max 8)**
**Three reviewers significantly increased their scores, all indicating the revised manuscript has adequately resolved their core technical concerns and raised meaningful future directions.**

---

### Meta-Review · Area_Chair_Pndw · 2026-01-06

**Summary:**

This paper presents an agentic toolchain that maps STEM images to simulation-ready structures (CIF) and downstream property prediction, and introduces a synthetic benchmark. Reviewers agree the end-to-end "image > structure > property" framing is valuable, and the system-level integration is a meaningful contribution. However, multiple reviewers raised substantial concerns that make acceptance difficult in a single-pass cycle: (i) the agent contribution is perceived as orchestration over a largely fixed pipeline rather than a clearly novel methodology, (ii) positioning vs. prior agentic scientific frameworks and microscopy pipelines is initially insufficient and risks overstating novelty, (iii) empirical grounding is limited by predominantly simulation-based evaluation with minimal real experimental validation, (iv) fairness/interpretability of comparisons and metric attribution are unclear in places (e.g., baselines not designed for the full loop; energy MAE conflating MLIP bias and reconstruction error), and (v) dependence on template retrieval and dataset/tier counting definitions raise questions about robustness and benchmark clarity.

**Reviewer Concerns:**

The authors have tried to address by the rebuttal and revision the followings:
* Clarified the role of the LLM or VLM as a tool orchestrator rather than a visual reasoner, reducing claim ambiguity about multimodal understanding
* Explained the hybrid control design with explicit threshold gating plus LLM re-planning, retries, and rollbacks, partially alleviating concerns that the workflow is a purely scripted linear pipeline
* Improved positioning relative to related systems and tightened the narrative around what is new in the end-to-end loop
* Corrected benchmark definition issues by clarifying case versus instance counting and tier construction
* Added ablations and controller swap analyses that better separate toolchain effects from controller choice, and clarified the scope of module-level baselines
* Added a limited real iDPC-STEM case study and a human-effort comparison, improving empirical grounding and practical framing

While the rebuttal and revisions resolve several clarity concerns, the extent of the changes required (+ the remaining gaps, particularly broader validation on real-world images and clearer methodological positioning) suggest that the paper would benefit from a more comprehensive revision and resubmission. Given the conference's single-round review process, I do not think the manuscript can reasonably reach a publishable standard within the current cycle.

**Reviewer Scores:**

Except the reviewer mXyq, all the other reviewers (including the ones who have not responded for the second round) asked a lot of concerns that require another round of revisions.

---

### Decision · Program_Chairs · 2026-01-26

Reject